**EMBO** *reports*

# Proteolytic cleavage of G3BP1 by calpain 1 couples NMDAR activation to mTOR-dependent local translation

Da-ha Park[1,2,3], So-Young Ahn [iD] [1,2,3], Jungho Kim[1,2,3], Jiwoo Choi [iD] [1,2,3], Seungha Lee[1,2,3], Minji Kang[1,2,3], Jae-man Song[1,2,3] & Young Ho Suh [iD] [1,2,3]✉

## Abstract

**Ribonucleoprotein (RNP) granules are dynamic, membraneless organelles that sequester translationally repressed mRNAs and RNA-binding proteins, playing a pivotal role in the regulation of localized protein synthesis. While disassembly of RNP granules is essential for reactivating translation, the mechanisms by which neuronal activity regulates this process remain poorly understood. In this study, we show that stimulation of N-methyl-D-aspartate (NMDA) receptor (NMDAR) triggers calcium influx, leading to activation of calpain 1 and subsequent proteolytic cleavage of Ras-GTPase-activating protein binding protein 1 (G3BP1), a core component of stress granules. This cleavage results in the disassembly of G3BP1 granules in the neurites and promotes mTOR-dependent local translation, thereby linking synaptic activity to spatially restricted protein synthesis. Finally, we demonstrate that the NMDAR–calpain 1–G3BP1–mTOR signaling axis contributes to axonal regeneration, establishing proteolytic remodeling of RNP granules as a key mechanism of activity-dependent neural repair.**

**Subject Categories** Neuroscience; Translation & Protein Quality

## Introduction

Local protein synthesis plays a pivotal role in establishing and maintaining synaptic plasticity, axonal regeneration, and neuronal circuit remodeling. Translationally repressed mRNAs are packaged into ribonucleoprotein (RNP) granules (also called RNA granules) and transported to distal neuronal compartments such as dendrites and axons, where they can be locally translated in response to specific environmental stimuli (Broix et al, 2021; Krichevsky and Kosik, 2001). The spatiotemporal regulation of RNP granules enables highly precise control of protein synthesis in both time and space, supporting activity-dependent synaptic remodeling that is essential for neurodevelopment and synaptic plasticity (Bauer et al, 2023; Hafner et al, 2019; Kiebler and Bauer, 2024).

RNA-binding proteins (RBPs), which comprise heterogeneous protein constituents within RNP granules such as stress granules (SGs), specifically interact with their target mRNAs with high affinity and regulate mRNA localization and metabolism (Formicola et al, 2019; Hentze et al, 2018). Among neuronal RBPs, Ras-GTPase-activating protein binding protein 1 (G3BP1) plays a central role in the dynamic assembly and disassembly of SGs and is increasingly being recognized for its functions beyond the classical stress response, particularly within neuronal compartments (Sidibé et al, 2021). G3BP1 contains five distinct domains, including three intrinsically disordered regions (IDRs) (Kang et al, 2021). The N-terminal nuclear transport factor-like (NTF2L) domain mediates homodimerization, which is essential for liquid-liquid phase separation (LLPS) and SG assembly, as well as protein–protein interactions required for subcellular localization (Yang et al, 2020). The acidic-rich IDR1 negatively regulates LLPS and modulates G3BP1 conformation (Guillén-Boixet et al, 2020). Under stress conditions, conformational changes in the IDRs expose the RNA-binding domain (RBD), facilitating G3BP1 binding to RNA and promoting SG formation (Guillén-Boixet et al, 2020; Yang et al, 2020). The proline-rich IDR2 (PxxP) contains SH3 domain-binding motifs that potentially provide interaction sites for various signaling partners (Sidibé et al, 2021). The C-terminal RBD comprises an RNA recognition motif (RRM) and an arginine-glycine-rich (RG-rich) IDR3. While the RRM confers RNA-binding specificity and initiates SG formation, the RG-rich IDR3 enhances the affinity of G3BP1 for RNA and stabilizes the G3BP1-RNA complex (Guillén-Boixet et al, 2020).

N-methyl-D-aspartate (NMDA) receptors (NMDARs) are glutamate-gated ion channels that permit calcium influx into neurons, generating intracellular calcium transients. NMDAR-mediated calcium entry plays a central role in numerous physiological processes, including neuronal development, synapto-genesis, synapse-to-nucleus signaling, synaptic plasticity, and higher-order functions, such as learning, memory, and cognition (Lau et al, 2009). Although NMDARs are best known for their

[1]Department of Biomedical Sciences, Seoul National University College of Medicine, Seoul, South Korea. [2]Neuroscience Research Institute, Medical Research Center, Seoul National University, Seoul, South Korea. [3]Transplantation Research Institute, Medical Research Center, Seoul National University, Seoul, South Korea. ✉E-mail: suhyho@snu.ac.kr

postsynaptic functions, presynaptic NMDARs (preNMDARs) are also widely expressed at specific synapses throughout the central nervous system (CNS), where preNMDARs have been implicated in the regulation of neurotransmitter release and various forms of synaptic plasticity (Bouvier et al, 2015; Wong et al, 2021). PreNMDAR expression is particularly enriched during early postnatal development and often includes juvenile subunits such as GluN2B and GluN3A. The low $Mg^{2+}$ sensitivity of preNMDARs suggests that strong stimuli, such as high-frequency firing or axonal injury, may enhance preNMDAR activation and downstream signaling. Notably, a recent study demonstrated that high-frequency neurotransmission in neocortical layer 5 pyramidal neurons is sustained through a preNMDAR-mTOR signaling pathway that controls local protein synthesis (Wong et al, 2024).

NMDAR stimulation triggers the activation of calpain, a calcium-dependent cysteine protease. Particularly, calpain 1 activation downstream of NMDARs contributes to long-term potentiation (LTP) in the hippocampus through the proteolytic modification of synaptic scaffolding proteins and key signaling molecules (Baudry and Bi, 2016, 2025; Liu et al, 2008; Vinade et al, 2001). Additionally, under physiological conditions or during the early phases of excitotoxicity, calpain 1 has been shown to confer neuroprotection through pathways involving synaptic NMDARs (Chiu et al, 2005; Hardingham, 2009; Lankiewicz et al, 2000; Wang et al, 2013; Xu et al, 2009). Despite these advances, the mechanism by which calpain 1 regulates RNP granules to support site-specific proteome remodeling in response to neuronal activity remains largely unclear.

In this study, we demonstrated that activation of the NMDAR–calpain 1 axis leads to the proteolytic cleavage of G3BP1, specifically at the proline-rich IDR2. We found that calpain 1-induced G3BP1 cleavage disassembles G3BP1-positive granules and promotes mTOR-dependent local translation in neurites. This NMDAR–calpain 1–G3BP1–mTOR signaling axis contributes to axonal regeneration in an injury model. Our findings provide mechanistic insights into the neuroprotective roles of NMDAR–calpain signaling by linking local RNP granule dynamics to activity-dependent translational control.

## Results

### Glutamate stimulation induces G3BP1 truncation and disassembly of G3BP1 granules

To investigate the activity-dependent dynamics of G3BP1-positive RNP granules, we stimulated days in vitro (DIV) 16 rat primary hippocampal neurons with potassium chloride (KCl)-induced depolarization, L-glutamate, or sodium arsenite-induced oxidative stress for 2 h. Although the calculated molecular weight of G3BP1 is approximately 52 kDa, the observed band in neurons appears at approximately 70 kDa, likely due to unique structural and biochemical features of G3BP1, such as its acidic domain and IDRs, which may alter its migration during SDS-PAGE (Fig. 1A). We found that both KCl and L-glutamate treatments produced approximately 50 kDa truncated form of G3BP1 in the Triton X-100-soluble (Tx-sol) fraction; however, this truncated G3BP1 fragment was barely detectable in the Triton X-100-insoluble (Tx-insol) fraction (Fig. 1A,B). In contrast, sodium arsenite

treatment did not affect G3BP1 truncation in either fraction (Fig. 1A,B). Notably, L-glutamate treatment markedly reduced the expression level of full-length G3BP1 in the Tx-insol fraction (Fig. 1A,B). These findings suggest that the truncated form of G3BP1 becomes detergent-soluble and is redistributed into the Tx-sol fraction.

Immunostaining of G3BP1 in neurons revealed that L-glutamate treatment did not cause notable changes in G3BP1 signals in the soma, whereas sodium arsenite strongly induced SG formation (Fig. 1C). In contrast, L-glutamate treatment significantly reduced the intensity of G3BP1-positive puncta in both dendrites and axons (Fig. 1D–H), suggesting that neuronal activation promotes G3BP1 truncation and disassembly of G3BP1 granules.

### NMDAR activation induces truncation and disassembly of G3BP1-positive granules

Since both depolarization and glutamate receptor stimulation induced G3BP1 truncation, we hypothesized that ionotropic glutamate receptors might mediate G3BP1 truncation. To test this hypothesis, we first examined whether selective agonists of NMDARs, AMPA receptors, or group I metabotropic glutamate receptors (mGluRs) induce G3BP1 truncation. Among these agonists, 50 µM NMDA treatment for 2 h robustly induced truncation of G3BP1 in DIV 16 primary hippocampal neurons, whereas AMPA or DHPG treatment had no obvious effect (Fig. 2A). Next, to investigate whether other mGluRs contribute to G3BP1 truncation, we applied a panel of pharmacological inhibitors targeting specific mGluR subtypes: MCPG (group I and II), CPCCOEt (mGluR1), LY367385 (mGluR1), MPEP (mGluR5), LY341495 (group II), MMPIP (mGluR7), and MSOP (group III). None of these antagonists prevented G3BP1 truncation (Fig. 2B), further confirming that G3BP1 truncation specifically occurs upon NMDAR activation.

To determine whether the effect of NMDA treatment is directly dependent on NMDAR channel activity, we treated primary hippocampal neurons with D-AP5, a competitive NMDAR antagonist, or MK-801, an open-channel blocker. Both inhibitors completely abolished NMDA-induced G3BP1 truncation (Fig. 2C), indicating that NMDAR signaling through activated NMDAR channels is essential for G3BP1 truncation. Additionally, NMDA treatment resulted in the redistribution of truncated G3BP1 into the Tx-sol fraction, accompanied by a decrease in the Tx-insol fraction (Fig. 2D), suggesting the disassembly of G3BP1-positive granules, which is consistent with the truncation effects observed following L-glutamate stimulation.

Since we observed two bands for native G3BP1, we hypothesized that the upper band might reflect post-translational modifications (PTMs). To test this possibility, we treated neuronal lysates with lambda phosphatase for up to 24 h to remove phosphate groups from serine, threonine, and tyrosine residues. However, the two bands remained unchanged following treatment (Fig. EV1), suggesting that the upper band is not derived from phosphorylation-dependent PTMs. These findings raise the possibility that the two bands represent alternative transcript isoforms of G3BP1. Additionally, since the intensities of both bands were similarly reduced following NMDA treatment, they appear to be equally susceptible to proteolytic cleavage. We next asked whether G3BP1 truncation could be observed in neurons stimulated with a

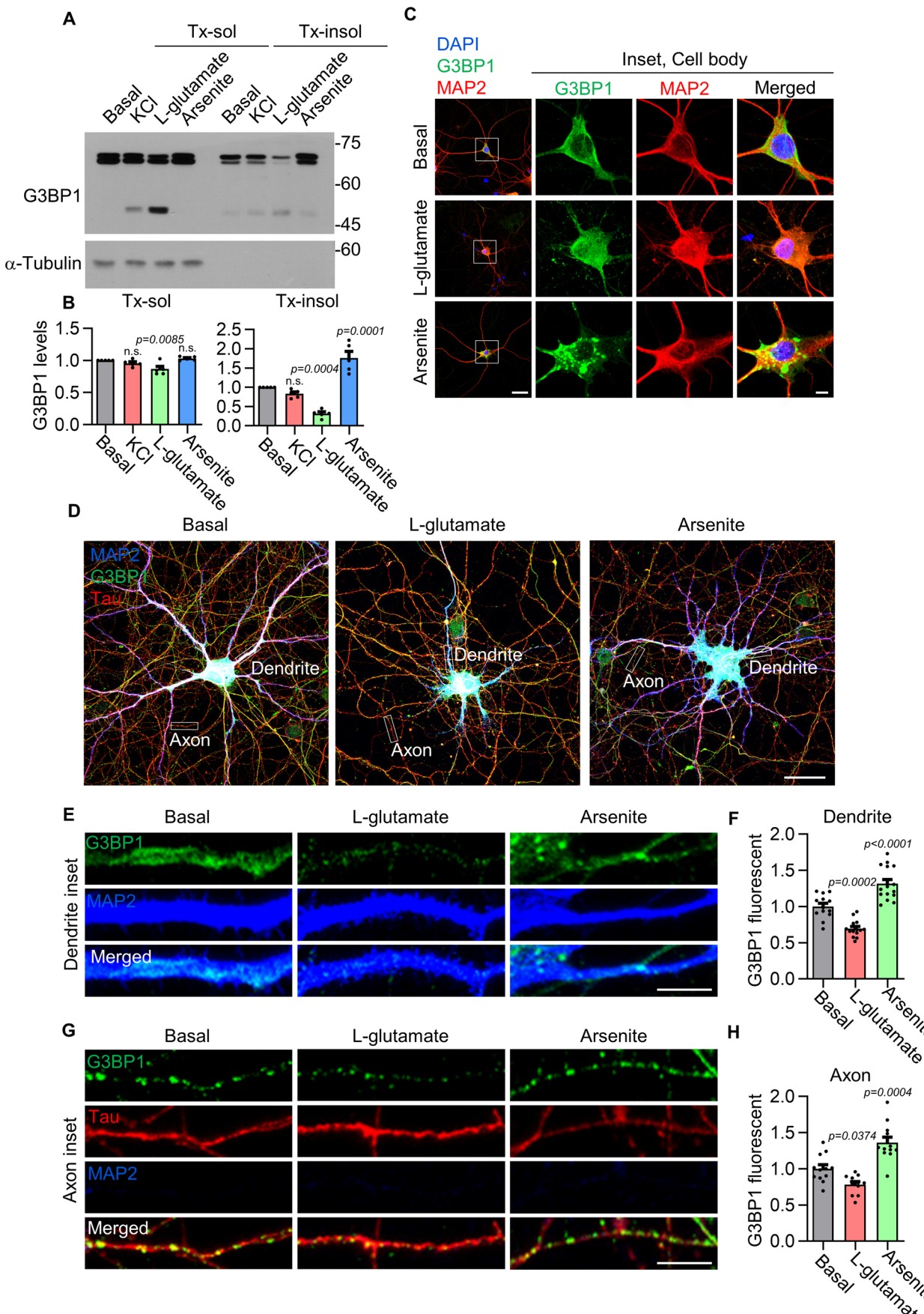

**Figure 1. Glutamate induces G3BP1 truncation and granule disassembly.**

(A) Western blot showing G3BP1 truncation in Triton X-100 soluble (Tx-sol) and insoluble soluble (Tx-insol) fractions from DIV 16 primary hippocampal neurons after 2 h treatment with KCl (50 mM), ʟ-glutamate (100 µM), or sodium arsenite (200 µM). (B) Quantification of full-length G3BP1 levels in each fraction. Data are shown as mean ± SEM (Tx-sol: $n = 5$, n.s., $P = 0.6107$ for KCl and $P = 0.8368$ for arsenite; Tx-insol, $n = 5$, n.s., $P = 0.5773$; all by one-way ANOVA with Tukey's post hoc test). (C) Confocal images of the soma of DIV 16 primary hippocampal neurons. Neurons were immunostained for G3BP1 and MAP2 after treatment with ʟ-glutamate or sodium arsenite. Scale bar, 10 µm. (D–H) G3BP1 granule disassembly in dendrites and axons following ʟ-glutamate or sodium arsenite treatment. Inset images in (E, G) are taken from (D). Quantification of G3BP1 signal intensity in MAP2-positive dendrites (F; $n = 3$, number of dendrites = 13–16; one-way ANOVA with Tukey's post hoc test) and Tau-positive axons (H; $n = 3$, number of axons = 11–12; one-way ANOVA with Tukey's post hoc test). Data are presented as mean ± SEM. Scale bars: 30 µm (D), 5 µm (E, G). Source data are available online for this figure.

protocol known to induce AMPA receptor endocytosis and chemical long-term depression without causing excitotoxicity (Beattie et al, 2000). When primary hippocampal neurons were exposed to a 3-min NMDA pulse, G3BP1 was truncated after 2 h, at levels comparable to those induced by 2 h of NMDA treatment (Fig. 2E). Furthermore, co-treatment with the extracellular calcium chelator EGTA prevented NMDA-induced G3BP1 truncation (Fig. 2F). These results indicate that transient calcium influx through NMDAR channels is sufficient to trigger G3BP1 truncation. Similar to L-glutamate treatment, NMDA stimulation reduced the intensity of G3BP1-positive puncta in both MAP2-positive dendrites and Tau-positive axons (Fig. 2G–J), suggesting that G3BP1 truncation is accompanied by disassembly of G3BP1 granules.

To exclude the possibility that the truncated form of G3BP1 might arise from de novo synthesis via alternative splicing following NMDA stimulation, we treated neurons with translation inhibitors, including cycloheximide (CHX), emetine, and puromycin. None of these inhibitors prevented G3BP1 truncation (Fig. 2K), supporting that the truncated form is not a product of translation but rather of proteolytic cleavage. Finally, we evaluated whether degradation pathways contribute to G3BP1 cleavage. Treatment with the proteasome-specific inhibitor epoxomicin or the autophagolysosome inhibitor bafilomycin A1 (Baf-A1) did not prevent NMDA-induced G3BP1 truncation (Fig. 2L). However, treatment with MG-132, a proteasome inhibitor that also exhibits inhibitory activity against cysteine proteases such as calpains and cathepsins (Kisselev and Goldberg, 2001; Tsubuki et al, 1996), robustly suppressed the production of truncated G3BP1 (Fig. 2L). These results suggest that the truncated form is generated by specific proteolytic cleavage, rather than by partial degradation.

## NMDA stimulation cleaves G3BP1 through the action of calcium-activated calpain 1

Given that G3BP1 cleavage depends on intracellular calcium and occurs via proteolytic cleavage, we focused on calpain, a well-characterized calcium-activated cysteine protease. Indeed, treatment with MDL-28170, a selective calpain inhibitor, completely blocked NMDA-induced G3BP1 cleavage in DIV 16 primary hippocampal neurons (Fig. 3A). To further investigate the role of calpain in G3BP1 cleavage, we performed the calpain cleavage assay. Following co-transfection of HEK 293T cells with G3BP1 and calpain constructs, cell lysates were prepared and incubated with 2 mM CaCl₂ to activate calpains. We found that calpain 1 cleaved endogenous G3BP1, whereas calpain 2 had little effect (Fig. 3B). A similar pattern was observed upon G3BP1

overexpression (EGFP-G3BP1), with calpain 1 promoting cleavage while calpain 2 showed minimal activity (Fig. 3C).

To determine calpain 1 cleavage sites in G3BP1, we generated a dual-tagged G3BP1 construct containing EGFP at the N-terminus and a Myc epitope at the C-terminus. In the in vitro cleavage assay, two distinct bands as N-terminal fragments were detected at ~65–70 kDa, and the C-terminal fragment appeared at approximately 25 kDa (Fig. 3D). The presence of multiple truncated bands suggests that G3BP1 may expose additional calpain cleavage sites under in vitro assay conditions. To confirm that calpain-mediated cleavage occurs in cells, we co-transfected HEK 293T cells with EGFP-G3BP1-Myc and FLAG-calpain 1, and then treated the cells with 1 µM ionomycin, a calcium ionophore, in the presence of 5 mM CaCl₂ for 30 min. Under these conditions, we observed robust G3BP1 cleavage specifically mediated by calpain 1 expression (Fig. 3E,F).

## Calpain 1 constitutively binds to G3BP1 and cleaves it at the proline-rich IDR2

To identify the calpain-sensitive region within G3BP1, we generated a series of deletion mutants lacking one of the four major domains: the NTF2L domain, acidic IDR1, proline-rich IDR2 (PxxP), or RBD (Fig. 4A). Using these mutants, we found that calpain 1 cleaved G3BP1 within the proline-rich IDR2 (PxxP) domain in both the in vitro calpain cleavage assay (Fig. 4B) and the calcium-ionomycin assay (Fig. 4C,D).

Next, we examined whether calpain 1 interacts with G3BP1. Co-immunoprecipitation using anti-calpain 1 antibody in primary hippocampal neuronal lysates showed a clear constitutive interaction between endogenous calpain 1 and G3BP1 (Fig. 4E). In addition, we observed substantial co-localization of endogenous calpain 1 and G3BP1 in axons by confocal microscopy (Fig. 4F). Co-transfection of HEK 293T cells with G3BP1 and calpain 1 followed by co-immunoprecipitation revealed robust binding between the two proteins (Fig. 4G). Notably, their interaction was unaffected under calcium-chelated conditions with EDTA and EGTA (Fig. 4G), indicating that calpain 1-G3BP1 binding is constitutive and calcium independent. To identify G3BP1 domains required for interaction with calpain 1, we analyzed a series of domain-deletion mutants. We found that calpain 1 binding was reduced in G3BP1 mutants lacking either the NTF2L domain or the acidic IDR1 region (Fig. 4H). We further generated G3BP1 constructs lacking either the NTF2L-acidic domains or the PxxP-RBD domains (Fig. 4A) and confirmed that calpain 1 interacted with the NTF2L and acidic domains (Fig. 4I). In vitro calpain cleavage assay revealed reduced cleavage in the mutant lacking the

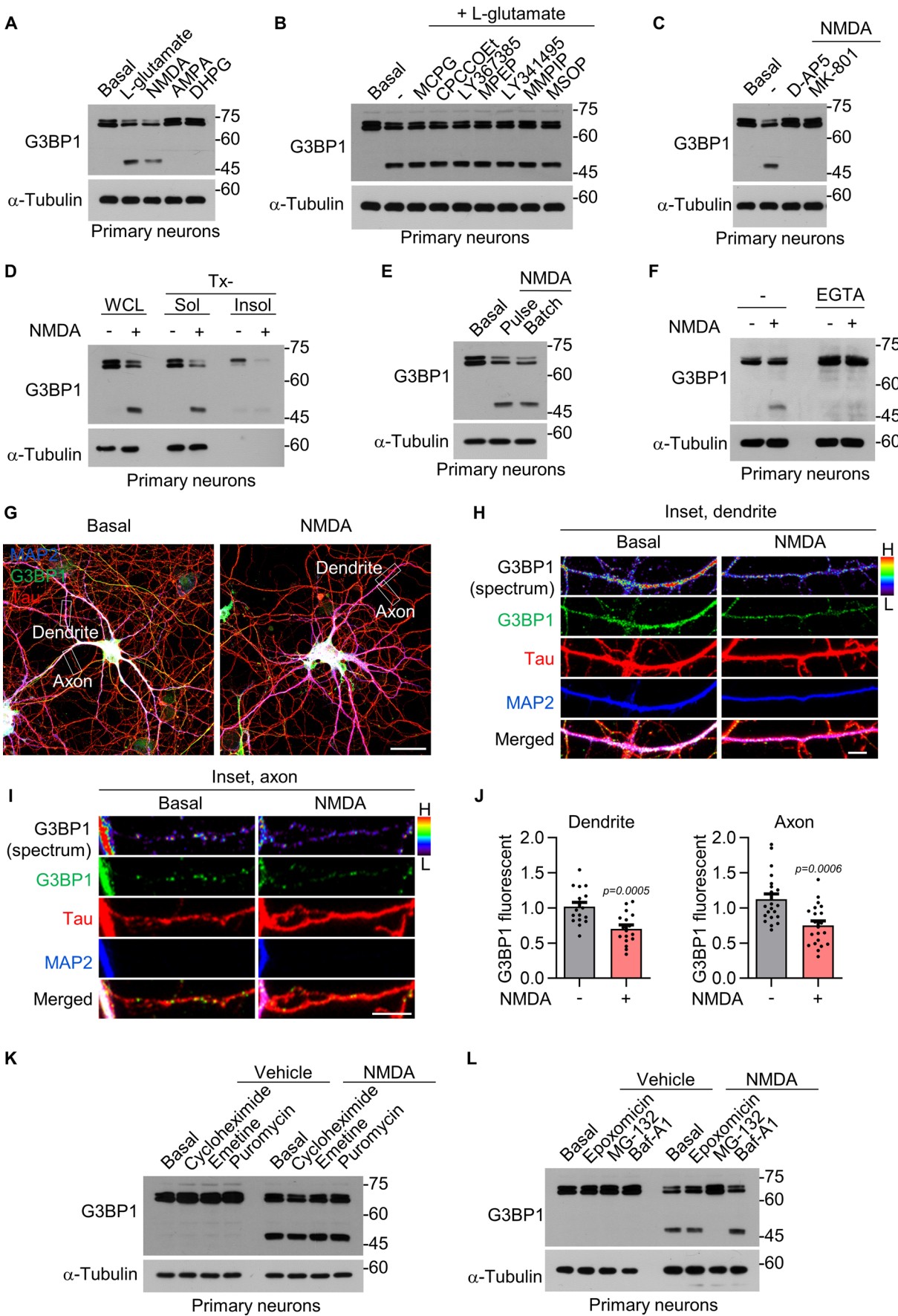

**Figure 2. NMDAR activation and calcium influx induce G3BP1 truncation and granule disassembly.**

(A) G3BP1 truncation after 2 h treatment with NMDA (50 µM), AMPA (100 µM), DHPG (100 µM), or ʟ-glutamate (100 µM). (B) Effect of mGluR inhibitors on ʟ-glutamate-induced G3BP1 truncation. (C) G3BP1 truncation after co-treatment with D-AP5 (100 µM) or MK-801 (10 µM) and NMDA. (D) Redistribution of G3BP1 into the Tx-sol fraction following NMDA treatment. (E) G3BP1 truncation after brief (3-min pulse + 2 h) or prolonged (2 h) NMDA exposure. (F) Effect of calcium chelator EGTA (50 mM) on NMDA-induced G3BP1 truncation. (G–I) Confocal images showing granule disassembly in dendrites and axons after NMDA treatment. DIV 16 primary hippocampal neurons were immunostained for G3BP1, MAP2, and Tau. Inset images in (H, I) are taken from (G). (J) Quantification of G3BP1 signal intensity. Data are shown as mean ± SEM (dendrites: $n = 3$, number of dendrites = 17; axons: $n = 3$, number of axons = 21; each by Student's $t$ test). Scale bars: 30 µm (G), 5 µm (H, I). (K, L) Western blot analysis of NMDA-induced G3BP1 truncation in the presence of translation inhibitors (K) or proteasome/autophagolysosome inhibitors (L). Source data are available online for this figure.

NTF2L-acidic domains and no detectable cleavage in the mutant lacking the PxxP-RBD domains (Fig. 4J). These results suggest that constitutive docking of calpain 1 to the NTF2L and acidic regions facilitates activity-dependent cleavage of G3BP1 at the proline-rich IDR2 domain. Under basal conditions, calpain 1 constitutively associates with the NTF2L and acidic regions of G3BP1 without exhibiting proteolytic activity. Upon NMDAR-mediated calcium influx, calpain 1 undergoes a conformational change that activates its proteolytic triad, enabling the cleavage of G3BP1 at the proline-rich IDR2. This cleavage event separates the NTF2L and RBD domains of G3BP1 (Fig. 4K).

## Calpain 1-mediated G3BP1 cleavage promotes disassembly of G3BP1 granules and enhances local translation

G3BP1 functions as a SG nucleator and translational repressor (Ivanov et al, 2019; Panas et al, 2016; Sahoo et al, 2018; Yang et al, 2020). Given that NMDA stimulation led to a reduction in full-length G3BP1 levels within the Tx-insol fraction and promoted the redistribution of truncated G3BP1 into the Tx-sol fraction (Fig. 2D), we examined whether calpain 1 activity regulates these changes and consequently disrupts G3BP1 function. We found that treatment with MDL-28170 prevented NMDA-induced G3BP1 truncation and redistribution in primary hippocampal neurons, thereby blocking NMDA-induced decrease in full-length G3BP1 levels in the Tx-insol fraction (Fig. 5A). In contrast, NMDA treatment increased TDP43 levels in the Tx-insol fraction, and this effect was unaffected by calpain 1 inhibition (Fig. 5A).

We investigated whether the observed decrease in insoluble G3BP1 following NMDA stimulation is associated with neuronal toxicity, we compared three conditions treated for 2 h: NMDA (which induces granule disassembly), NMDA with calpain inhibition (which prevents disassembly), and sodium arsenite (which promotes granule assembly). Using an LDH cytotoxicity assay, we found no significant differences in neuronal viability across these conditions (Fig. EV2). These results suggest that, under the conditions used in our study, neither NMDA-induced granule disassembly nor arsenite-induced granule formation exerts overtly beneficial or detrimental effects on neuronal viability.

We next examined activity-dependent changes in G3BP1-positive granules following NMDA stimulation. NMDA treatment led to a marked reduction in the intensity of G3BP1-positive puncta in axons of primary hippocampal neurons, and this effect was blocked upon co-treatment with the calpain inhibitor MDL-28170 (Fig. 5B,C). These results indicate that calpain 1 activity promotes the disassembly of G3BP1-positive granules, including SGs.

## The NMDAR–calpain 1 pathway regulates G3BP1 granule-associated local translation via mTOR

Since G3BP1-positive SGs contain translationally repressed transcripts and associated ribosomes, we investigated whether calpain 1-mediated disassembly of G3BP1 granules regulates protein synthesis. To address this, primary hippocampal neurons were incubated with 10 µM puromycin for 30 min to perform puromycin-based surface sensing of translation (SUnSET) assay (Schmidt et al, 2009). Puromycin is a tyrosyl-tRNA analog that is incorporated into nascent peptides, enabling the detection of newly synthesized proteins. The puromycin labeling assay showed that NMDA stimulation markedly reduced global protein synthesis; however, this reduction was not reversed by MDL-28170 treatment (Fig. 5D). These findings suggest that global translation is independent of the NMDAR–calpain signaling pathway and likely occurs primarily in the soma.

To determine whether the NMDAR–calpain pathway regulates local translation in neurites, we performed the ribopuromycylation (RPM) assay. In this assay, emetine prevents the release of puromycylated nascent peptides from translating ribosomes, thereby allowing the detection of actively translating mRNAs within ribosomal complexes. NMDA stimulation significantly increased puromycin incorporation near G3BP1 granules in axons, whereas this NMDA-induced increase in local translation was abolished by MDL-28170 treatment (Fig. 5E,F), indicating derepression of local protein synthesis within G3BP1-positive granules.

We further examined whether calpain 1-mediated cleavage induced by NMDA is associated with local translation using a cleavage-resistant G3BP1 ΔPxxP mutant. In neurons expressing the ΔPxxP mutant, NMDA-induced increases in the RPM signal were not observed in neurons expressing the ΔPxxP mutant (Fig. 5G,H). Taken together, these results support a model in which calpain 1-mediated G3BP1 cleavage relieves translational repression and disrupts the function of G3BP1 as a local translational repressor.

We next investigated the molecular mechanism by which the NMDAR–calpain 1–G3BP1 signaling axis regulates local translation. It has been well established that the mammalian target of rapamycin (mTOR) phosphorylates eukaryotic initiation factor 4E-binding proteins (4E-BPs), thereby preventing their association with eIF4E and facilitating the assembly of translation initiation complex on dendritic and axonal mRNAs (Panas et al, 2016; Wolozin and Ivanov, 2019). Notably, mTOR has been shown to interact with G3BP1 (Prentzell et al, 2021; Schwarz et al, 2015) and is activated downstream of NMDAR signaling to promote dendritic and axonal protein synthesis (Gong et al, 2006; Wong et al, 2024). Thus, we investigated whether mTOR is involved in local translation regulated by the NMDAR–calpain 1–G3BP1 axis.

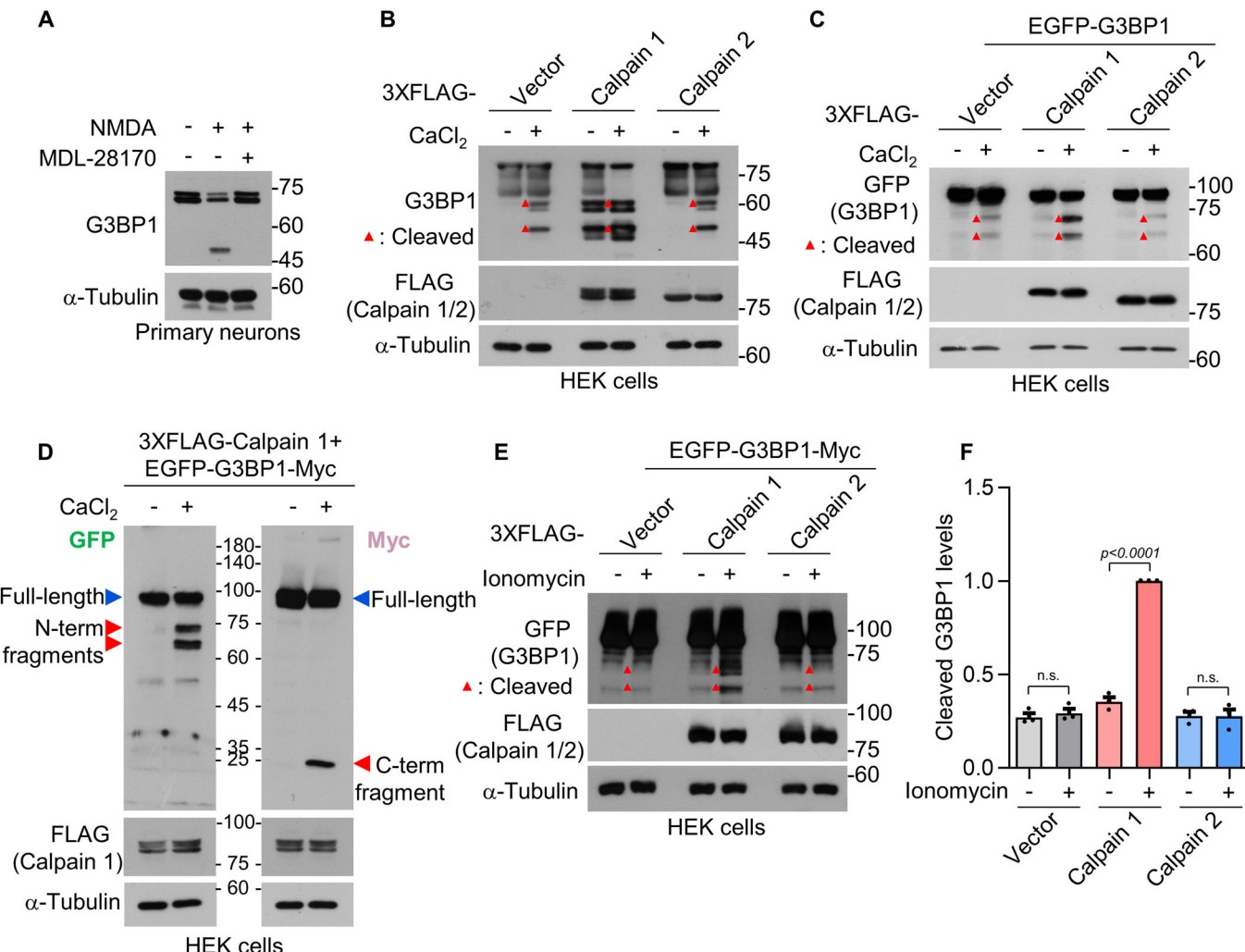

**Figure 3. Calpain 1 cleaves G3BP1 following NMDAR stimulation.**

(A) Western blot analysis of G3BP1 cleavage after NMDA treatment with or without the calpain inhibitor MDL-28170 (5 μM) for 2 h. (B, C) In vitro calpain cleavage assay using HEK 293T cell lysates expressing FLAG-calpain 1 or 2 with or without calcium chloride (50 μM). The expression of endogenous G3BP1 (B) or exogenous EGFP-tagged G3BP1 (C) was analyzed by western blotting. (D) Western blot showing calpain 1-generated cleavage fragments detected using anti-GFP and anti-Myc antibodies, which recognize the N-terminal and C-terminal regions, respectively. (E) Western blot showing G3BP1 cleavage following calcium-ionomycin treatment (5 mM $CaCl_2$, 1 μM ionomycin, 30 min) in HEK 293T cells co-transfected with FLAG-calpain 1 and EGFP-G3BP-Myc. (F) Quantification of cleaved G3BP1 levels relative to full-length G3BP1, normalized to the calpain 1 + ionomycin condition. Data are shown as mean ± SEM ($n = 3$, n.s., $P = 0.9901$ for vector and $P > 0.9999$ for calpain 2; one-way ANOVA with Tukey's post hoc test). Source data are available online for this figure.

Towards this, we used in situ puromycin proximity ligation assay (puro-PLA), a technique that visualizes nascent protein synthesis at subcellular resolution (De Pace et al, 2025; tom Dieck et al, 2015). We observed an increase in mTOR translation in the axons of primary hippocampal neurons following NMDA stimulation. This NMDA-induced axonal mTOR translation was diminished by MDL-28170 treatment (Fig. 5I,J), indicating a critical role for calpain 1 in activity-dependent local translation through mTOR.

To further confirm that NMDAR–calpain 1 signaling regulates local translation, we cultured primary hippocampal neurons on porous membrane inserts using a Boyden chamber system, which physically separates neurites from cell bodies. We then performed RPM assays in the presence of emetine and puromycin. NMDA treatment increased local translation in the neurite-enriched fraction, and this effect was reversed by co-treatment with MDL-28170 (Fig. 5K,L). These results further support that NMDA-induced local protein synthesis occurs in a calpain-dependent manner within neurites. In contrast, western blot analyses of phosphorylated mTOR, p70S6K, and 4E-BP1 in neurite-enriched fractions did not reveal a measurable increase in mTOR activity (Figs. EV3A,B). We propose that this may reflect the presence of a substantial pool of pre-existing mTOR protein, which could obscure subtle changes in newly synthesized or activated mTOR. While Puro-PLA imaging can selectively detect nascent mTOR synthesis, bulk western blotting lacks the spatial and temporal resolution to distinguish these localized, activity-dependent changes from the pre-existing mTOR pool.

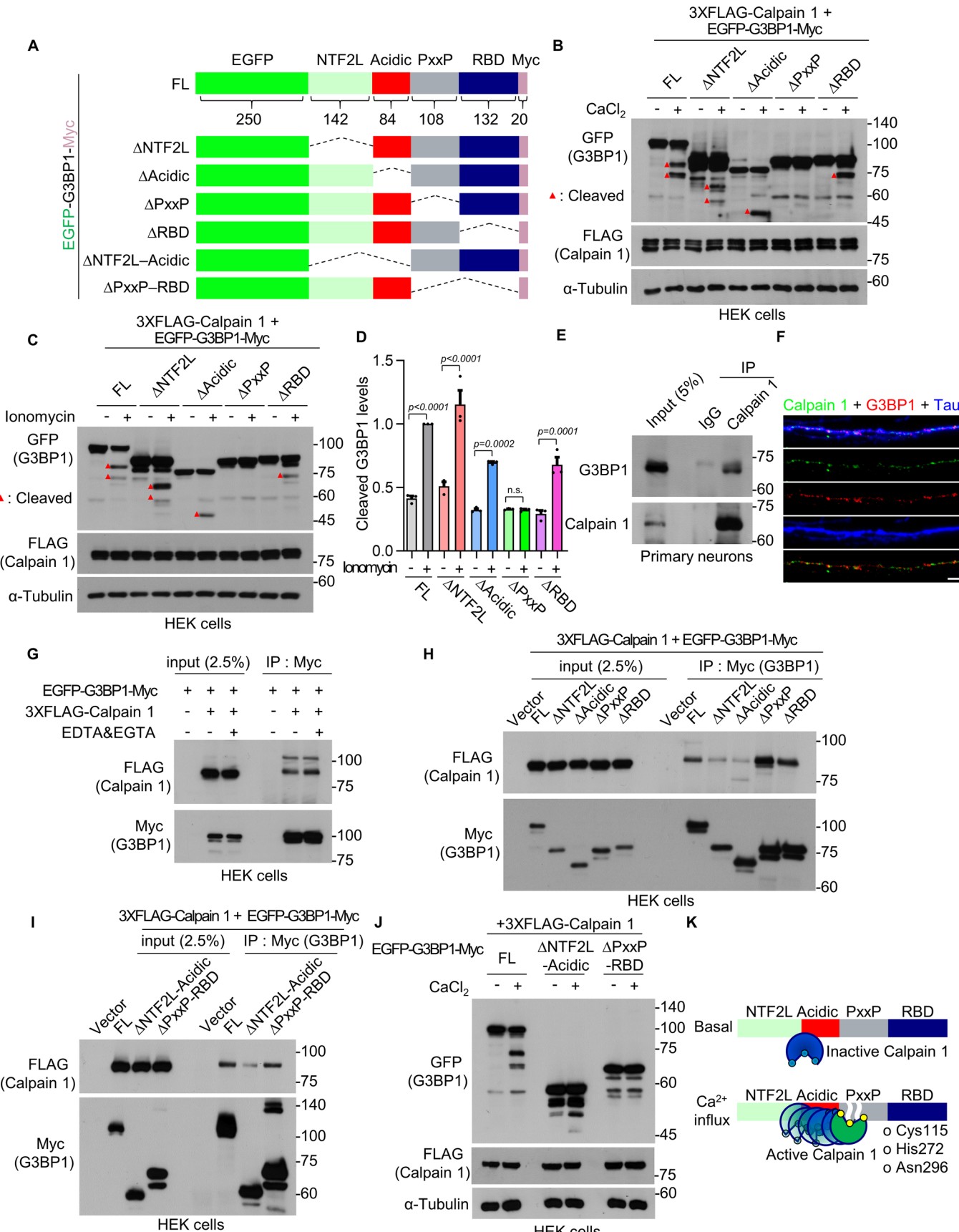

**Figure 4. Calpain 1 cleaves the PxxP domain of G3BP1 and constitutively interacts with the NTF2L and acidic domains.**

(A) Schematic diagram of EGFP-G3BP1-Myc mutant constructs lacking the indicated domains. FL, full-length. The numbers represent the number of amino acids. (B) Western blot showing in vitro calpain cleavage assay in G3BP1 mutants lacking the indicated domain. (C) Western blot analysis of G3BP1 mutant cleavage by calpain 1 following calcium-ionomycin treatment. (D) Quantification of cleaved G3BP1 levels normalized to the full-length + ionomycin condition. Data are shown as mean ± SEM (n = 3, n.s., P > 0.9999; one-way ANOVA with Tukey's post hoc test). (E) Co-immunoprecipitation assay using anti-calpain 1 antibody from primary hippocampal neuron lysates. (F) Confocal images showing co-localization of calpain 1 and G3BP1 in distal axons. Scale bar, 5 μm. (G) HEK 293T cell lysates, co-expressing EGFP-G3BP1-Myc and FLAG-calpain 1, with or without EDTA and EGTA, were used to perform co-immunoprecipitation using anti-Myc antibody. (H, I) Domain-specific interactions between G3BP1 mutants and calpain 1 were analyzed by co-immunoprecipitation using anti-Myc antibody. (J) In vitro calpain cleavage assay in G3BP1 mutants lacking the indicated domain. (K) Model depicting the proteolytic cleavage of G3BP1 by basal and activated calpain 1. Source data are available online for this figure.

## The NMDAR–calpain 1–G3BP1–mTOR axis regulates axonal regeneration

It has been reported that locally translated mTOR regulates axonal local translation following nerve injury (Terenzio et al, 2018). Therefore, we investigated whether calpain 1-mediated upregulation of mTOR contributes to axonal regeneration. To test this hypothesis, we performed the scrape-induced axonal regeneration assay using primary hippocampal neurons. Following mechanical injury generated by 40 intersecting scratches with a sterile 10 μL micropipette tip, we observed a gradual increase in G3BP1 cleavage beginning at 1 h post-injury (Fig. 6A,B). These results suggest that the scrape assay could be a suitable model for evaluating calpain 1-mediated G3BP1 cleavage characterized in this study. Moreover, we found that blocking NMDARs with MK-801 or D-AP5 suppressed G3BP1 cleavage in the scrape injury model (Fig. 6C,D). Furthermore, treatment with MDL-28170 prevented G3BP1 cleavage (Fig. 6E), suggesting that the NMDAR–calpain 1 signaling axis mediates G3BP1 cleavage in this injury context.

To evaluate axonal regeneration, we immunostained the neurons with the beta-III tubulin antibody 48 h after scraping (Fig. 6F,G). Axonal regrowth into the scraped area was markedly inhibited by treatment with MDL-28170, Torin 1, or the calcium chelator EGTA. These results indicate that calcium-activated calpain 1 and mTOR signaling are essential for axonal regeneration. In particular, quantification of axonal regeneration showed that Torin 1 treatment resulted in an ~60% reduction in regrowth compared to the control (Fig. 6G), confirming that mTOR plays a significant role in the regenerative process.

To determine isoform-specific contributions of calpain to injury-induced axonal regeneration, we knocked down calpain 2 in neurons and found that calpain 2 inhibition alone did not significantly affect axonal regrowth (Figs. EV4A–C). In contrast, treatment with MDL-28170, which inhibits both calpain 1 and calpain 2, markedly reduced axonal regeneration under conditions of calpain 2 knockdown (Figs. EV4A–C), indicating that the regenerative process is primarily dependent on calpain 1 activity.

We next examined the functional consequences of blocking G3BP1 cleavage by expressing a calpain-resistant G3BP1 ΔPxxP mutant in G3BP1 knockdown neurons. Expression of the ΔPxxP mutant significantly reduced axonal regeneration (Fig. 6H,I), consistent with the notion that intact G3BP1 granules suppress regenerative capacity. Taken together, these findings support a model in which the NMDAR–calpain 1–G3BP1–mTOR signaling axis plays a critical role in promoting axonal regeneration following injury.

## Discussion

In this study, we demonstrated that neuronal activity, specifically through NMDAR activation, triggers calcium-dependent activation of calpain 1 and induces calpain-mediated proteolytic cleavage of G3BP1, leading to the disassembly of G3BP1-containing RNP granules. This process enhances local translation and provides the proteome necessary for axonal regeneration in an mTOR-dependent manner. In contrast to oxidative stress, NMDAR stimulation induces calpain 1-dependent truncation of G3BP1 and its redistribution into the Tx-sol fraction. This selective proteolytic event does not affect global translation, but instead modulates local translation in neurites, revealing a calcium-dependent mechanism that supports activity-dependent, spatially restricted protein synthesis at the subcellular level. Furthermore, we uncovered a previously unrecognized post-translational regulatory mechanism that links neuronal activity to proteolytic remodeling of RNP granules.

Our findings establish G3BP1 as a proteolytic substrate of calpain 1, with its proline-rich IDR2 domain serving as the cleavage site. Previous studies have shown that the positively charged RG-rich IDR3 is critical for LLPS and SG formation through RNA binding, whereas the negatively charged acidic-rich IDR1 functions as an autoinhibitory domain that suppresses granule assembly (Guillén-Boixet et al, 2020; Yang et al, 2020). Under basal conditions, G3BP1 adopts a closed conformation through intra-molecular interactions between IDR1 and IDR3. Upon binding of RNA to IDR3, the G3BP1 homodimer undergoes a conformational transition to an open state, facilitating its assembly into SG condensates. Importantly, the positively charged IDR2 interacts with the negatively charged IDR1 in a charge-dependent manner, thereby interfering with the formation of the IDR1-IDR3 interface. Therefore, calpain 1-mediated cleavage within IDR2 possibly reinforces the closed conformation of G3BP1 by promoting the engagement of IDR1 with IDR3, ultimately preventing granule assembly. Moreover, cleavage of IDR2 separates the NTF2L domain from the RBD region, which likely destabilizes the overall structural integrity of G3BP1 granules. The calpain-cleaved N-terminal fragment of G3BP1 likely retains dimerization capacity with full-length G3BP1, which could interfere with LLPS and stress granule formation. While the C-terminal fragment may retain RNA-binding activity, it appears to be highly unstable and was difficult to detect in neurons.

Notably, this proteolytic event does not appear to be stochastic, but rather represents a programmed and tightly regulated mechanism. We found that calpain 1 constitutively binds to the

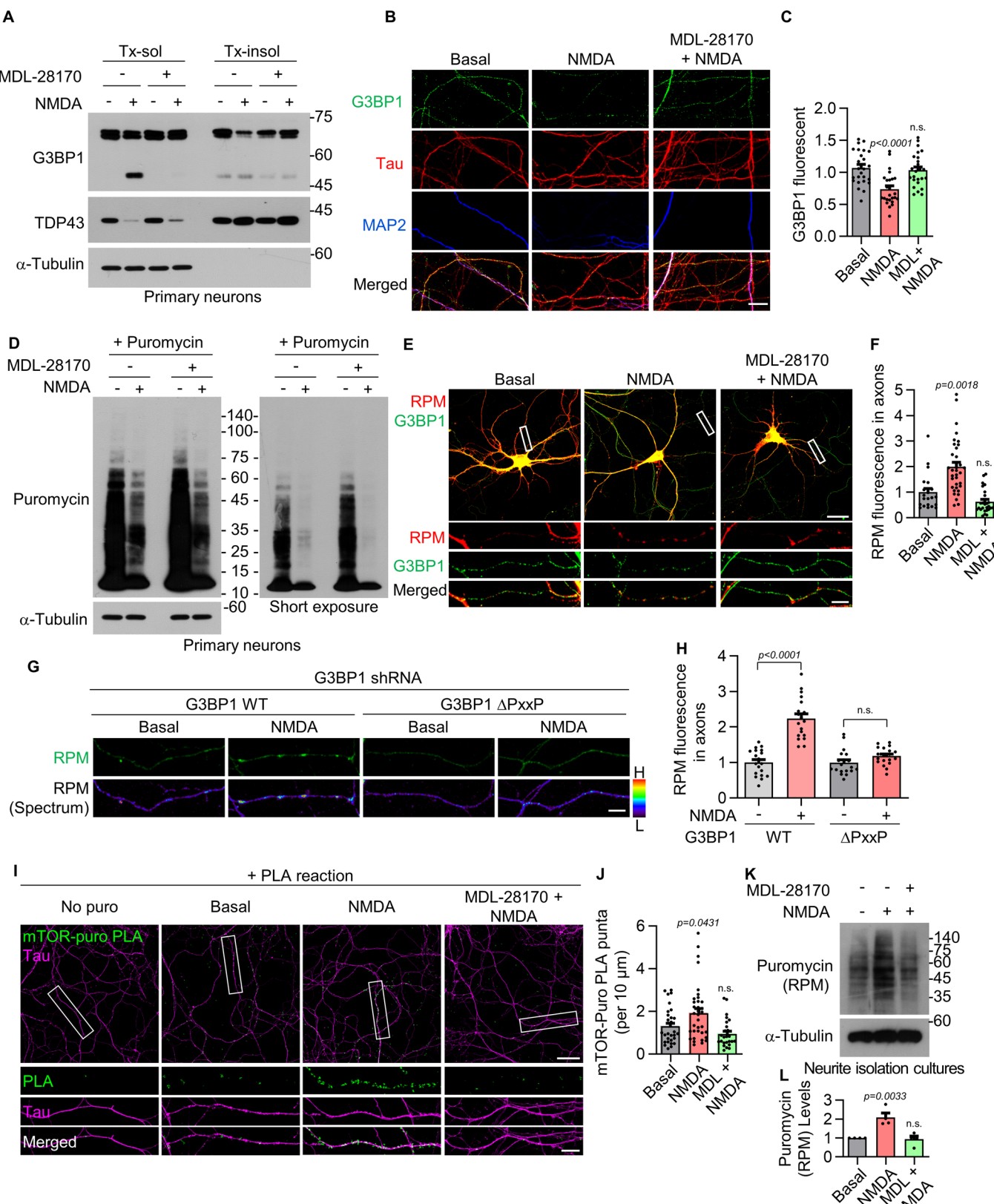

◄ **Figure 5. Calpain 1-mediated G3BP1 cleavage and granule disassembly promote local translation.**

(A) Western blot showing that MDL-28170 prevents NMDA-induced redistribution of G3BP1 into the Tx-sol fraction. (B) Confocal images showing the effect of MDL-28170 on G3BP1 granules in Tau-positive, MAP2-negative axons of DIV 14 primary hippocampal neurons. Scale bar, 10 µm. (C) Quantification of axonal G3BP1 signal intensity from (B) is shown as mean ± SEM ($n = 3$, number of axons = 25, n.s., $P = 0.8682$; one-way ANOVA with Tukey's post hoc test). (D) Western blot analysis of puromycin-labeled nascent polypeptide synthesis following NMDA and/or MDL-28170 treatment. The right panel shows a short exposure of the same blot. (E) Confocal images of ribopuromycylation (RPM) assay showing active translation near G3BP1 granules in axons. Scale bar, 30 µm; inset region, 5 µm. (F) Quantification of RPM signal intensity under each treatment condition in (E) is shown as mean ± SEM ($n = 3$, number of axons = 22–34, n.s., $P = 0.1384$; Kruskal–Wallis test followed by Dunn's post hoc test). (G) Confocal images of RPM assay. Corresponding spectrum images represent intensity heatmaps of puromycin signal distribution. (H) Quantification of RPM signal intensity under each condition in (G) is shown as mean ± SEM ($n = 3$, number of axons = 19, n.s., $P = 0.4856$; one-way ANOVA with Tukey's post hoc test). (I) In situ puromycin proximity ligation assay (puro-PLA) to detect nascent mTOR translation (mTOR-puro-PLA) in Tau-positive axons. Scale bar, 30 µm; inset region, 10 µm. (J) Quantification of axonal mTOR-puro-PLA signal intensity from panel (I) is shown as mean ± SEM ($n = 3$, number of axons = 25–35, n.s., $P = 0.4118$; one-way ANOVA with Tukey's post hoc test). (K) Western blot analysis of puromycin incorporation (RPM signals) in neurite-enriched fractions in the presence of emetine under the indicated conditions. (L) Quantification of RPM signals from (K) is shown as mean ± SEM ($n = 4$, n.s., $P = 0.9568$; one-way ANOVA with Tukey's post hoc test). Source data are available online for this figure.

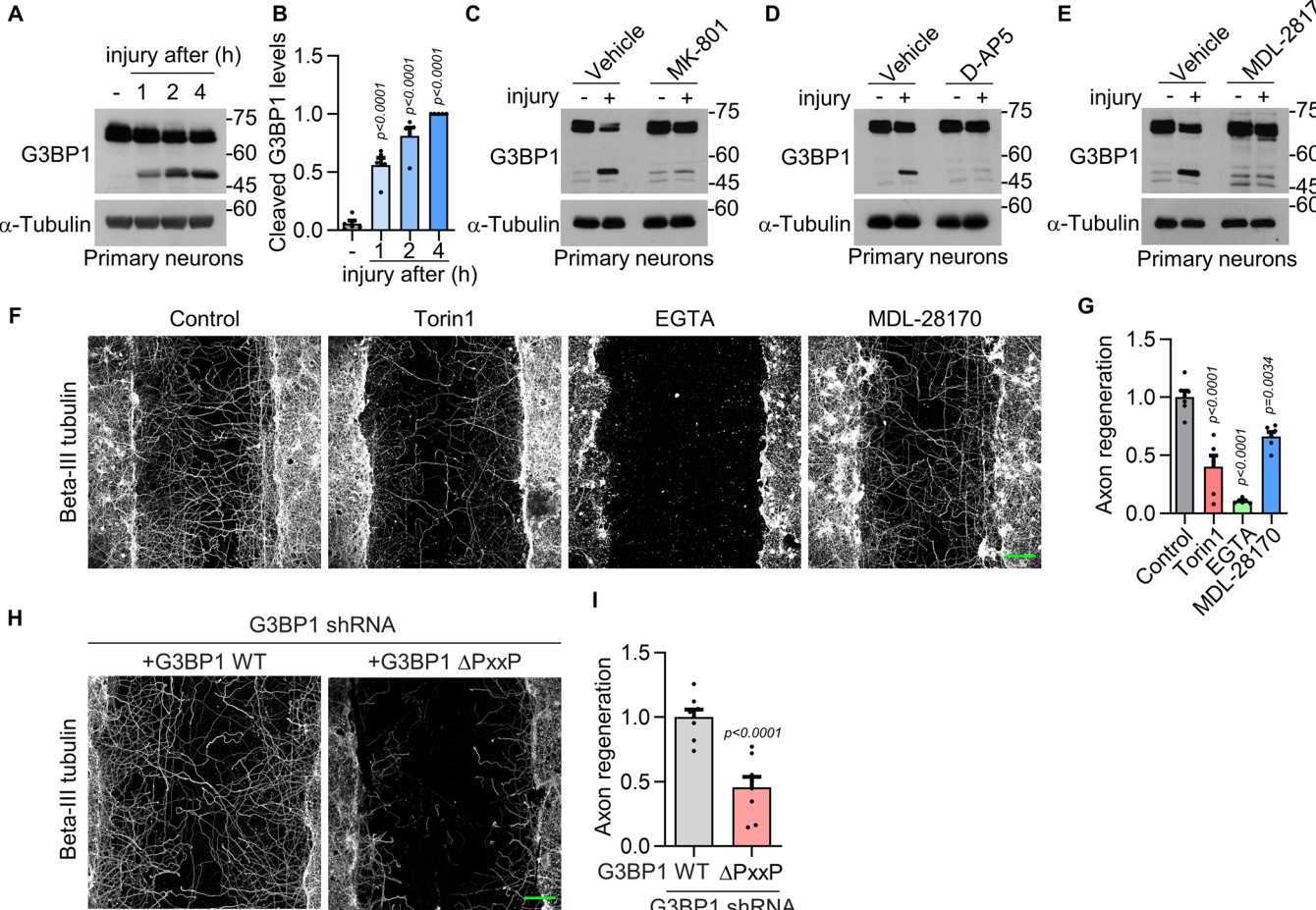

**Figure 6. NMDAR–calpain 1 signaling supports axonal regeneration after injury.**

(A) Western blot showing time-course of G3BP1 cleavage after scrape injury of DIV 10–14 primary hippocampal neurons. (B) Quantification of cleaved G3BP1 levels normalized to the 4 h post-injury condition from (A). Data are shown as mean ± SEM ($n = 5$; one-way ANOVA with Tukey's post hoc test). (C, D) Inhibition of G3BP1 cleavage by NMDAR antagonists MK-801 (C) and D-AP5 (D) following injury. (E) Effect of calpain inhibitor MDL-28170 on G3BP1 cleavage in the scrape injury model. (F) Confocal images showing axonal regeneration visualized by beta-III tubulin immunostaining under the indicated treatment conditions. Scale bar, 100 µm. (G) Quantification of axonal regeneration from (F) is shown as mean ± SEM ($n = 6$; one-way ANOVA with Tukey's post hoc test). (H) Confocal images showing axonal regeneration in neurons expressing either G3BP1 WT or the calpain 1-resistant mutant ΔPxxP under endogenous G3BP1 knockdown. Scale bar, 100 µm. (I) Quantification of axonal regeneration from (H) is shown as mean ± SEM ($n = 8$; Student's t test). Source data are available online for this figure.

NTF2L and acidic domains of G3BP1 under resting conditions and remains inactive. Upon calcium influx through NMDAR activation, calpain 1 becomes catalytically active and cleaves G3BP1 at IDR2 (Fig. 4K). These observations support a model in which G3BP1 is pre-configured for rapid, stimulus-responsive cleavage via a calcium-dependent mechanism. Such spatially and temporally gated proteolysis enables the dynamic regulation of G3BP1 granules and facilitates local translation in response to synaptic activity.

Previous studies have demonstrated that SG components can act as negative regulators of axonal regeneration by repressing local protein synthesis through SG-like aggregation. For example, phosphorylation of G3BP1 at Ser149 promotes SG disassembly and enhances local translation of axonal mRNAs (Sahoo et al, 2025; Sahoo et al, 2018). Disrupting G3BP1-mediated granule formation has also been shown to activate axonal translation and accelerate regeneration (Sahoo et al, 2025; Sahoo et al, 2018). Similarly, TIAR-2 facilitates LLPS-driven granule formation and inhibits axon regeneration in *C. elegans* (Andrusiak et al, 2019). These findings align with our results and support a model in which proteolytic disassembly of G3BP1 granules is essential for facilitating the axonal regeneration process.

We observed NMDA-induced G3BP1 cleavage in both the axons and dendrites. While classical NMDARs are predominantly localized to postsynaptic dendrites, preNMDARs are also expressed in axons, particularly during the early stages of synaptic development in the CNS (Banerjee et al, 2016; Bouvier et al, 2015). These preNMDARs play important roles in synapse formation, neurotransmitter release, and synaptic plasticity. Their synaptogenic nature may be linked to the local synthesis of proteomes via mTOR signaling and may contribute to neuronal regeneration following injury.

While tightly regulated NMDAR activity supports neuroprotection and cellular resilience, excessive or dysregulated calcium influx can lead to neuronal damage and cell death during brain injury and neurodegenerative disorders. Consequently, pharmacological blockade of NMDARs often exacerbates neuronal vulnerability, impairs recovery following injury, or renders neurons more susceptible to secondary insults (Hardingham, 2009). Indeed, clinical trials using NMDAR antagonists for traumatic brain injury and stroke have largely failed or produced detrimental outcomes. These observations have led to the proposal that the activation, rather than inhibition, of NMDARs during the subacute phase post-injury may be more beneficial (Ikonomidou and Turski, 2002; Shohami and Biegon, 2014). Notably, the two major calpain isoforms, calpain 1 and calpain 2, have been shown to play opposing roles in the brain. Calpain 1, which is activated via synaptic NMDARs, mediates neuroprotective signaling, whereas calpain 2, which is activated through extrasynaptic NMDARs, contributes to axonal degeneration and neuronal death (Adamec et al, 1998; Baudry, 2019; Baudry and Bi, 2016, 2025; Wang et al, 2013). These reports align with our findings that selective activation of calpain 1 through NMDARs cleaves G3BP1, but not calpain 2, resulting in axonal regeneration and neuroprotective effects. Isoform-specific outcomes may arise from differences in calcium sensitivity ($EC_{50}$), subcellular localization, scaffold associations, substrate preferences, and downstream signaling pathways (Averna et al, 2015; Baudry and Bi, 2016; Doshi and Lynch, 2009; Metwally et al, 2023; Xu et al, 2009). Furthermore, given that the NMDAR

subunits, GluN2A and GluN2B, are calpain substrates (Bi et al, 1998; Guttmann et al, 2001; Monnerie et al, 2010; Simpkins et al, 2003; Tompa et al, 2004; Vinade et al, 2001; Wu et al, 2005), a negative feedback regulatory loop may exist within the NMDAR–calpain signaling axis. Taken together, our findings provide mechanistic insights into a model in which the NMDAR–calpain 1–G3BP1–mTOR axis functions as a pathway that supplies locally synthesized proteomes to support synaptic transmission and neuronal regeneration. Nonetheless, as the current study is based on primary cultured neuronal models, future investigations using in vivo injury models will be essential to further elucidate the physiological relevance and functional significance of the signaling axis in neural repair.

We demonstrated that NMDARs specifically activate calpain 1 via calcium influx. Since depolarization by potassium chloride or calcium-ionophore treatment in heterologous systems can also activate calpain 1, we cannot rule out the possibility that other calcium-permeable channels, such as transient receptor potential (TRP) channels or voltage-gated calcium channels (VGCCs), may contribute to calpain activation under certain conditions (Kanamori et al, 2013; Kerstein et al, 2013). Furthermore, it would be highly informative to perform a mass spectrometry-based approach to identify the full spectrum of calpain 1 substrates following NMDAR activation. For example, a recent study successfully mapped calpain 2 substrates (Kapilan et al, 2025), and similar efforts targeting calpain 1 could provide invaluable insights into the broader proteolytic signaling network governed by calpain 1. Nonetheless, our findings support a model wherein the NMDAR–calpain 1–G3BP1–mTOR signaling axis constitutes a key activity-dependent mechanism that promotes axonal regeneration, thereby offering a promising therapeutic target for CNS repair.

# Methods

**Reagents and tools table**

| Reagent/resource | Reference or source | Identifier or catalog number |
|---|---|---|
| **Experimental models** | | |
| Primary cultured neuron | This study | N/A |
| HEK 293T | ATCC | Cat#CRL-3216 |
| **Recombinant DNA** | | |
| EGFPc2_G3BP1 | This study | N/A |
| EGFPc2-G3BP1-myc | This study | N/A |
| EGFPc2-G3BP1-ΔNTF2L-myc | This study | N/A |
| EGFPc2-G3BP1-ΔAcidic-myc | This study | N/A |
| EGFPc2-G3BP1-ΔPxxP-myc | This study | N/A |
| EGFPc2-G3BP1-ΔRBD-myc | This study | N/A |
| EGFPc2-G3BP1-ΔNTF2L-Acidic-myc | This study | N/A |
| EGFPc2-G3BP1-ΔPxxP-RBD-myc | This study | N/A |
| p3XFLAG-CAPN1 | Addgene | Cat# 60941 |
| p3XFLAG-CAPN2 | Addgene | Cat# 60942 |

| Reagent/resource | Reference or source | Identifier or catalog number |
|---|---|---|
| FHUGW vector | This study | N/A |
| FHUGW-rG3BP1 shRNA | This study | N/A |
| FUW-G3BP1-myc | This study | N/A |
| FUW-G3BP1-ΔPxxP-myc | This study | N/A |
| FHUGW-rCAPN2 shRNA #1 | This study | N/A |
| FHUGW-rCAPN2 shRNA #2 | This study | N/A |
| pCMV-VSV-G | Addgene | Cat# 8454 |
| Delta 8.9 | This study | N/A |
| **Antibodies** | | |
| Rabbit anti-G3BP1 | Proteintech | Cat# 13057-2-AP |
| Mouse anti-α tubulin | Sigma | Cat# T6199 |
| Rabbit anti-FLAG | Sigma | Cat# F7425 |
| Mouse anti-GFP | Invitrogen | Cat# A11122 |
| Mouse anti-c-Myc | in house | N/A |
| Rabbit anti-TDP43 | Abcam | Cat# EPR5810 |
| Mouse anti-puromycin | Millipore | Cat# MABE343 |
| Mouse anti-MAP2 | Synaptic Systems | Cat# 188011 |
| Guinea pig anti-Tau | Synaptic Systems | Cat# 314 004 |
| Mouse anti-beta-III tubulin | Biolegend | Cat# 801201 |
| Mouse anti-CAPN1 | Proteintech | Cat# 67732-I-Ig |
| Mouse anti-CAPN2 | Proteintech | Cat# 66977-I-Ig |
| Rabbit anti-mTOR | Cell Signaling | Cat# 2972 |
| Rabbit anti-p-mTOR (Ser2448) | Cell Signaling | Cat# 2971 |
| Rabbit anti-p-mTOR (Ser2481) | Cell Signaling | Cat# 2974 |
| Rabbit anti-P70S6K | Cell Signaling | Cat# 9202 |
| Rabbit anti-p-P70S6K (Thr389) | Cell Signaling | Cat# 9234 |
| Rabbit anti-4E-BP | Cell Signaling | Cat# 9644 |
| Rabbit anti-p-4E-BP (Ser65) | Cell Signaling | Cat# 9451 |
| Rabbit anti-p-4E-BP (Thr37, 46) | Cell Signaling | Cat# 2855 |
| Peroxidase AffiniPure Goat Anti-Rabbit IgG | JacksonImmunoResearch | Cat# 111-035-144 |
| Peroxidase AffiniPure Goat Anti-Mouse IgG | JacksonImmunoResearch | Cat# 115-035-146 |
| Goat anti-Rabbit IgG (H + L) Highly Cross-Adsorbed Secondary Antibody, Alexa Fluor™ 488 | Invitrogen | Cat# A-11034 |
| Goat anti-Mouse IgG (H + L) Highly Cross-Adsorbed Secondary Antibody, Alexa Fluor™ 647 | Invitrogen | Cat# A-21236 |
| Goat anti-Mouse IgG (H + L) Highly Cross-Adsorbed Secondary Antibody, Alexa Fluor™ 568 | Invitrogen | Cat# A-11031 |

| Reagent/resource | Reference or source | Identifier or catalog number |
|---|---|---|
| Goat anti-Guinea Pig IgG (H + L) Highly Cross-Adsorbed Secondary Antibody, Alexa Fluor™ 647 | Invitrogen | Cat# A-21450 |
| **Oligonucleotides and other sequence-based reagents** | | |
| None | | |
| **Chemicals, enzymes and other reagents** | | |
| DMEM | welgene | Cat# LM001-05 |
| FBS | GenDEPOT | Cat# F0900-050 |
| Hanks' Balanced Salt Solution | Thermo Fisher Scientific | Cat# 14170-161 |
| trypsin | Sigma-Aldrich | Cat# T1005 |
| HEPES | Sigma-Aldrich | Cat# H3375 |
| Deoxyribonuclease I | Sigma-Aldrich | Cat# D5025 |
| Penicillin–streptomycin | Thermo Fisher Scientific | Cat# 15070-063 |
| Serum-free neurobasal medium | Thermo Fisher Scientific | Cat# 21103-049 |
| B-27 | Thermo Fisher Scientific | Cat# 17504-044 |
| GlutaMAX™ Supplement | Thermo Fisher Scientific | Cat# 35050061 |
| KCl | Sigma | Cat# P5405 |
| ʟ-Glutamate | Tocris | Cat# 0218 |
| Sodium arsenite | Sigma | Cat# 1062771000 |
| NMDA | Tocris | Cat# 0114 |
| AMPA | Tocris | Cat# 0254 |
| DHPG | Tocris | Cat# 0342 |
| MCPG | Tocris | Cat# 0336 |
| CPCCOEt | Tocris | Cat# 1028 |
| LY367385 | Tocris | Cat# 1237 |
| MPEP | Tocris | Cat# 1212 |
| LY341495 | Tocris | Cat# 1209 |
| MMPIP | Tocris | Cat# 2963 |
| MSOP | Tocris | Cat# 66515-29-5 |
| D-AP5 | Tocris | Cat# 0106 |
| MK-801 | Tocris | Cat# 0924 |
| EGTA | Amresco | Cat# 0732 |
| MDL-28170 | Merck Millipore | Cat# 208722 |
| jetOPTIMUS | Polyplus | Cat# 101000006 |
| Protein G Sepharose | Cytiva | Cat# 17061801 |
| EDTA | GenDEPOT | Cat# 11262412 |
| puromycin | Sigma | Cat# P8833 |
| emetine | Sigma | Cat# E2375 |
| Duolink® In Situ Red Starter Kit (Mouse/Rabbit) | Sigma | Cat# DUO92101 |
| TransIT-VirusGEN® Transfection Reagent | Mirus | MIR 6705 |
| Permeable Support for 6-well plate with 1.0 μm transparent PET membrane, Sterile | Falcon | 353102 |

| Reagent/resource | Reference or source | Identifier or catalog number |
|---|---|---|
| LDH-Cytotoxicity Colorimetric Assay Kit II | BIOVISON | K313-500 |
| **Software** | | |
| Leica Application Suite X | Leica-microsystems | RRID:SCR_013673 |
| GraphPad Prism 9 | GraphPad | RRID:SCR_002798 |
| **Other** | | |
| Leica STELLARIS 5 microscope | Leica-microsystems | RRID:SCR_024663 |
| Sunrise Microplate Reader | TECAN | RRID:SCR_027762 |

## Primary neuron culture

Primary hippocampal neurons were isolated from embryonic day 18 (E18) Sprague-Dawley rats obtained from OrientBio. All animal care and use procedures were performed in accordance with the guidelines established by the Seoul National University Institutional Animal Care and Use Committee (protocol no. SNU-240207-1-5). To minimize suffering, pregnant rats were euthanized by $CO_2$ asphyxiation followed by decapitation. Embryos were delivered via Cesarean section and immediately decapitated. Hippocampi were dissected and incubated in Hanks' Balanced Salt Solution containing 0.05% trypsin, 10 mM HEPES, 0.137 mg/mL deoxyribonuclease I, and penicillin–streptomycin at 37 °C for 12 min. Tissues were mechanically dissociated by trituration using a fire-polished Pasteur pipette. Cells were plated on poly-D-lysine-coated culture plates in serum-free Neurobasal medium supplemented with B-27 and 1% GlutaMAX™. Cultures were maintained in a humidified incubator at 37 °C and 5% $CO_2$, with half-medium changes every 2–3 days.

## Neurite isolation culture

Neurite isolation cultures were performed using cell culture inserts equipped with transparent polyethylene terephthalate (PET) membranes (pore size, 1.0 μm). Primary hippocampal neurons were plated onto the upper surface of the insert, allowing neurites to extend through the membrane pores to the lower surface, while somata remained in the upper compartment. This configuration enables physical separation of neurites from cell bodies for downstream biochemical analyses.

## Drug treatments and calcium modulation

To stimulate neuronal activity and intracellular calcium influx, neurons were treated for 2 h with 50 μM KCl, 100 μM L-glutamate, 200 μM sodium arsenite, 50 μM NMDA, 100 μM AMPA, or 100 μM DHPG. For receptor blockade experiments, mGluR antagonists (MCPG, 1 mM; CPCCOEt, 100 μM; LY367385, 10 μM; MPEP, 10 μM; LY341495, 5 nM; MMPIP, 1 μM; MSOP, 100 μM) or NMDAR antagonist (D-AP5, 50 μM; MK-801, 10 μM) were administered 5 min prior to NMDA treatment. Calcium chelation was achieved by adding 50 mM EGTA for 30 min prior to stimulation. Calpain activity was inhibited using 5 μM MDL-28170.

## Western blot analysis and subcellular fractionation

Cells were lysed in ice-cold TNE buffer (50 mM Tris-HCl, pH 7.5, 150 mM NaCl, 2 mM EDTA, and 1% Triton X-100) supplemented with protease and phosphatase inhibitor cocktails. Lysates were incubated on ice for 20 min and centrifuged at $14,000 \times g$ for 15 min at 4 °C to separate Triton X-100-soluble and -insoluble fractions. The insoluble pellets were resuspended in the same lysis buffer and sonicated. Western blotting was performed using antibodies against G3BP1, α-tubulin, FLAG, GFP, Myc, TDP-43, and puromycin, as appropriate. Protein bands were detected using enhanced chemiluminescence, exposed to X-ray film, and developed in a dark room using standard chemical developers and fixatives.

## Immunocytochemistry and confocal microscopy

Neurons were fixed with 4% paraformaldehyde containing 4% sucrose, and permeabilized with 0.25% Triton X-100. After blocking, the neurons were incubated with primary antibodies against G3BP1, MAP2, Tau, and puromycin. Alexa Fluor-conjugated secondary antibodies were used for detection. Confocal images were captured using the Leica STELLARIS 5 system. Regions of comparable thickness, as determined by MAP2- and Tau-positive immunostaining, were selected for analysis.

## In vitro cleavage assay

HEK 293T cells were transfected with EGFP-G3BP1-Myc and/or FLAG-calpain 1 or 2 using the jetOPTIMUS transfection reagent. Cleavage reactions were performed by mixing cell lysates with 50 μM $CaCl_2$ for 45 min at room temperature. The reactions were stopped by adding 6× Laemmli buffer and boiling prior to western blot analysis.

## Immunoprecipitation

For co-immunoprecipitation analysis, HEK 293T cells were lysed in TNE buffer. Lysates were incubated with anti-Myc antibody and Protein G Sepharose beads overnight at 4 °C. For calcium-sensitive interaction studies, 2 mM EDTA and 2.5 mM EGTA were added to the lysis and wash buffers. The bound proteins were eluted and analyzed by western blotting.

## Lactate dehydrogenase (LDH) cytotoxicity assay

Cytotoxicity of primary rat hippocampal neurons was assessed using the LDH-Cytotoxicity Colorimetric Assay Kit II (Biovision, #K313), according to the manufacturer's instructions. Briefly, neurons were treated with a 3-min pulse of 50 μM NMDA, followed by a 2-h incubation in conditioned medium with or without 5 μM MDL-28170, or treated with 200 μM sodium arsenite for 2 h. Untreated neurons were used as the low control to determine spontaneous LDH release, whereas 10% Cell Lysis Solution was added for 30 min to establish the high control. Culture supernatants were then collected, and 10 μL of each was mixed with 100 μL of LDH reaction mixture and incubated for 30 min at room temperature. Absorbance was measured at 450 nm using a microplate reader (TECAN). Cytotoxicity (%) was calculated by

normalizing to the low and high controls, following the manufacturer's protocol.

## SUnSET assay

To assess nascent protein synthesis (Schmidt et al, 2009), DIV 17 primary hippocampal neurons were treated with NMDA (50 μM) and/or MDL-28170 (5 μM), followed by a 30-min incubation with 10 μM puromycin at 37 °C. The cells were washed twice with ice-cold PBS, lysed in TNE buffer supplemented with 0.2% SDS, and analyzed by western blotting using anti-puromycin antibody.

## RPM assay

To label actively translating ribosomes, neurons were treated with 200 μM emetine for 15 min to arrest ribosomal protein synthesis, and then exposed to 100 μM puromycin for 10 min. After fixation, the cells were immunostained with anti-puromycin and anti-G3BP1 antibodies to visualize translating ribosomes in proximity to G3BP1 granules in the axons.

## mTOR-puro-PLA

To visualize nascent mTOR translation in situ, DIV 14 primary hippocampal neurons were incubated with 5 μM puromycin for 2 h. Where indicated, inhibitors were added 5 min prior to NMDA stimulation. Puro-PLA was performed using the Duolink® In Situ Red Starter Kit (Mouse/Rabbit) using anti-puromycin and anti-mTOR antibodies, according to the manufacturer's instructions with minor optimizations, including blocking with 10% normal goat serum to optimize conditions for primary neurons.

## Knockdown and overexpression of lentivirus

For gene knockdown, shRNA constructs were designed to target the following sequences: rat G3BP1, 5′-GGCTTGGATTCCAACGG-GAAG-3′; rat CAPN2 KD#1, 5′-GAGAGAGCCATCAAGTACC-3′; rat CAPN2 KD#2, 5′-GAGAAGAAGGCTGACTACC-3′. Lentiviruses for knockdown or overexpression were produced by co-transfecting HEK 293T cells with the lentiviral expression vector, packaging plasmid Δ8.9, and envelope plasmid pCMV-VSV-G using TransIT-VirusGEN® transfection reagent. Viral supernatants were collected and concentrated by ultracentrifugation at $110,000 \times g$ for 90 min, resuspended in PBS, and used for subsequent experiments.

## Mechanical injury model of neurons

Neurons were mechanically scraped as described previously (Mukhin et al, 1997; Tecoma et al, 1989). For biochemical analyses, DIV 10–14 primary hippocampal neurons were manually scraped using a sterile 10 μL micropipette tip to create intersecting horizontal and vertical lines across the culture dish, forming a grid with ~5 mm spacing between parallel lines. For immunostaining, single horizontal and vertical scratches were made across the culture dish.

## Quantification and statistical analysis

Quantitative image analysis (e.g., RPM intensity) was performed using ImageJ. Statistical analyses were performed using GraphPad

Prism or Microsoft Excel. No formal randomization or blinding was applied; however, all samples were processed simultaneously under standardized and controlled experimental conditions. For comparisons across multiple groups, one-way ANOVA followed by Tukey's post hoc test was used for normally distributed data, whereas the Kruskal–Wallis test followed by Dunn's post hoc test was used for non-normally distributed data. For comparisons between two groups Student's unpaired $t$ test was used. Data are presented as mean ± SEM from independent experiments. Sample sizes ($n$) and statistical tests are indicated in the corresponding figure legends. A $P$ value < 0.05 was considered statistically significant.

## Data availability

This study includes no data deposited in external repositories.

The source data of this paper are collected in the following database record: biostudies:S-SCDT-10_1038-S44319-026-00766-9.

## Peer review information

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

## Acknowledgements

This work was supported by grants from the National Research Foundation (NRF) of Korea (NRF-2020R1A5A1019023), the Korea Dementia Research Project through the Korea Dementia Research Center (KDRC) (RS-2024-00332875), and the Cooperative Research Program of Basic Medical Science and Clinical Science from Seoul National University College of Medicine and Seoul National University Hospital (800-20250051).

## Author contributions

**Da-ha Park**: Conceptualization; Resources; Data curation; Software; Formal analysis; Validation; Investigation; Visualization; Methodology. **So-Young Ahn**: Investigation; Methodology. **Jungho Kim**: Investigation; Methodology. **Jiwoo Choi**: Investigation; Methodology. **Seungha Lee**: Investigation; Methodology. **Minji Kang**: Investigation; Methodology. **Jae-man Song**: Investigation; Methodology. **Young Ho Suh**: Conceptualization; Resources; Data curation; Supervision; Funding acquisition; Writing—original draft; Project administration; Writing—review and editing.

Source data underlying figure panels in this paper may have individual authorship assigned. Where available, figure panel/source data authorship is listed in the following database record: biostudies:S-SCDT-10_1038-S44319-026-00766-9.

## Disclosure and competing interests statement

The authors declare no competing interests.

# Expanded View Figures

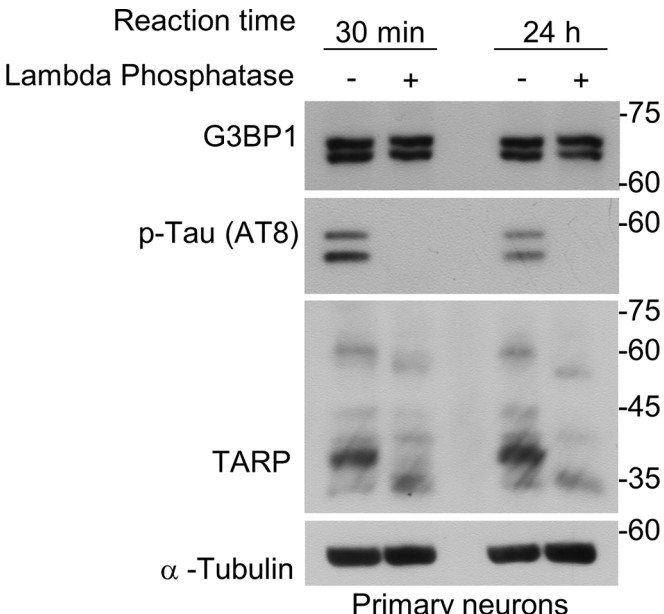

**Figure EV1.   The two bands of native G3BP1 are not due to phosphorylation-dependent post-translational modifications.**

Western blot analysis of primary hippocampal neuron lysates treated with λ-phosphatase for 30 min or 24 h. The double band pattern of G3BP1 persists following phosphatase treatment, indicating that the upper band is not derived from phosphorylation-dependent post-translational modifications. Source data are available online for this figure.

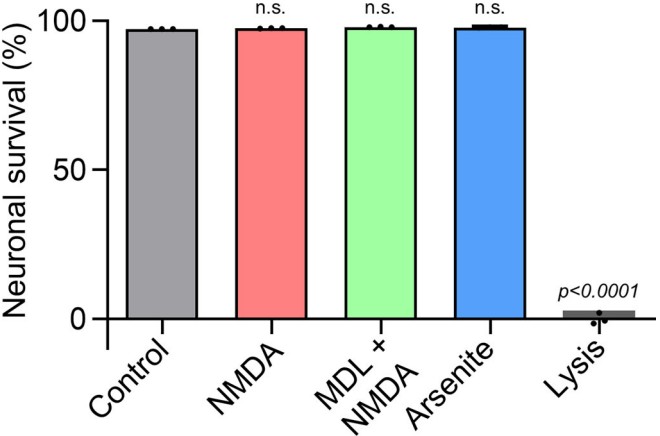

**Figure EV2. Granule assembly or disassembly conditions do not significantly affect neuronal survival.**

Neuronal survival was assessed by LDH cytotoxicity assay following 2-h treatments of DIV 14 primary hippocampal neurons. Conditions included granule disassembly (50 µM NMDA), prevention of disassembly (50 µM NMDA plus 5 µM MDL-28170), and granule assembly (200 µM sodium arsenite). No significant differences in cytotoxicity were observed among the conditions. Data are shown as mean ± SEM ($n = 3$, n.s., $P = 0.9910$ for NMDA, $P = 0.8722$ for MDL + NMDA, and $P = 0.9202$ for arsenite; one-way ANOVA with Tukey's post hoc test). Source data are available online for this figure.

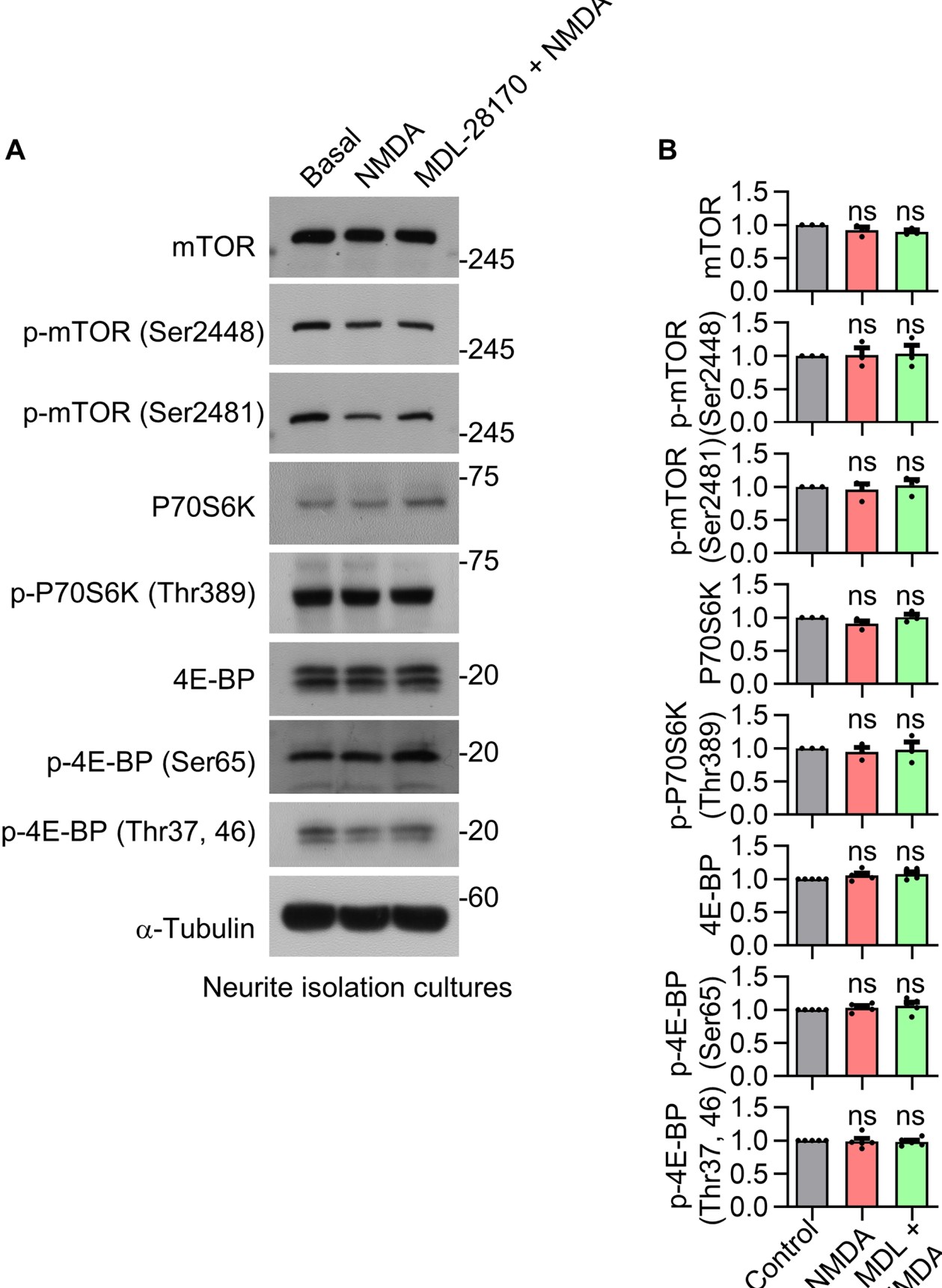

**A** Neurite isolation cultures

◀ **Figure EV3. Western blot analysis does not detect NMDA-induced changes in mTOR activity, even in neurite-enriched fractions.**

(A) Western blot analysis of mTOR downstream signaling in neurite-enriched fractions of primary hippocampal neurons. (B) Quantification of mTOR downstream signaling from (A) is shown as mean ± SEM ($n = 3$–5, n.s., $P > 0.05$; one-way ANOVA with Tukey's post hoc test). For exact $P$ values, please refer to the Source Data. Source data are available online for this figure.

**A**

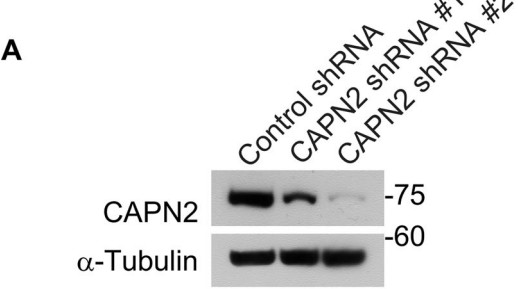

Control shRNA
CAPN2 shRNA #1
CAPN2 shRNA #2

CAPN2 -75

α-Tubulin -60

**B**

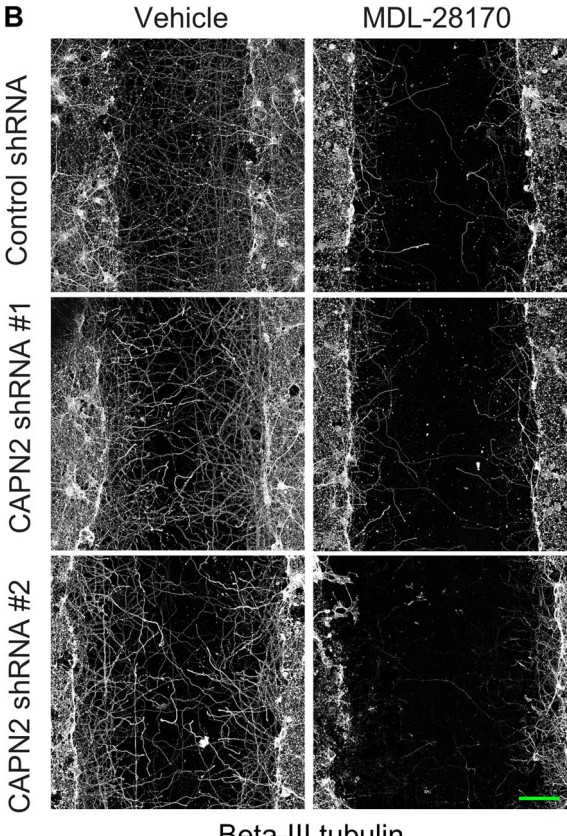

Vehicle MDL-28170

Control shRNA

CAPN2 shRNA #1

CAPN2 shRNA #2

Beta-III tubulin

**C**

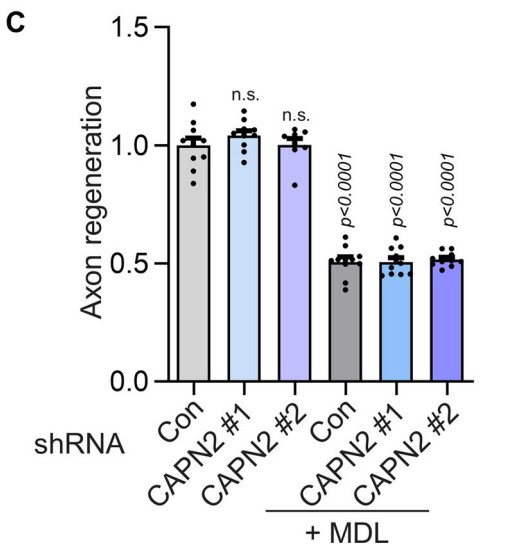

Axon regeneration

n.s. n.s.

$p<0.0001$ $p<0.0001$ $p<0.0001$

1.5

1.0

0.5

0.0

shRNA Con CAPN2 #1 CAPN2 #2 Con CAPN2 #1 CAPN2 #2

+ MDL

◀ **Figure EV4. Calpain 2 does not affect axonal regeneration.**

(A) Western blot showing the knockdown efficiency of calpain 2 (CAPN2) using shRNA in primary hippocampal neurons. (B) Representative confocal images showing axonal regeneration visualized by beta-III tubulin immunostaining in neurons expressing CAPN2 shRNA or treated with MDL-28170. Scale bar, 100 μm. (C) Quantification of axon regeneration from (B) is shown as mean ± SEM ($n = 8$–10, n.s., $P = 0.7330$ CAPN2 #1 and $P > 0.9999$ for CAPN2 #2; one-way ANOVA with Tukey's post hoc test). Source data are available online for this figure.

