## [Peer Review File · EMBO Reports]

Proteolytic cleavage of G3BP1 by calpain 1 couples NMDAR activation to mTOR-dependent local translation

Young Ho Suh, Da-ha Park, So-Young Ahn, Jungho Kim, Jiwoo Choi, Seungha Lee, Minji Kang, and Jae-man Song

Corresponding author(s): Young Ho Suh (suhyho@snu.ac.kr)

Review Timeline:

Submission Date:	22nd Sep 25
Editorial Decision:	30th Oct 25
Revision Received:	28th Jan 26
Editorial Decision:	9th Mar 26
Revision Received:	10th Mar 26
Accepted:	25th Mar 26

Editor: Esther Schnapp

Transaction Report:

Dear Prof. Suh,

Thank you for the submission of your manuscript to EMBO reports. We have now received the full set of referee reports that is pasted below.

As you will see, the referees acknowledge that the findings are potentially very interesting. However, together they also raise several concerns and have a number of suggestions for how the study should or could be improved. I think all suggestions are good and should be addressed. Especially referee 1 raises excellent points that all make perfect sense, but I am happy to discuss what kind of revisions you can do, as I understand that this referee is asking for a lot of data. I would like to suggest that you send me a proposed revision plan as point-by-point response that we can discuss in order to agree on a set of revisions. We can also do this in a video chat, if you like. Please let me know what you think.

I would thus in principle like to invite you to revise your manuscript with the understanding that the referee concerns must be fully addressed and their suggestions taken on board. Please address all referee concerns in a complete point-by-point response. Acceptance of the manuscript will depend on a positive outcome of a second round of review. It is EMBO reports policy to allow a single round of major revision only and acceptance or rejection of the manuscript will therefore depend on the completeness of your responses included in the next, final version of the manuscript.

We realize that it is difficult to revise to a specific deadline. In the interest of protecting the conceptual advance provided by the work, we recommend a revision within 3 months (30th Jan 2026). Please discuss the revision progress ahead of this time with the editor if you require more time to complete the revisions.

- 1) A data availability section providing access to data deposited in public databases is missing. If you have not deposited any data, please add a sentence to the data availability section that explains that.
- 2) Your manuscript contains statistics and error bars based on $n=2$. Please use scatter blots in these cases. No statistics should be calculated if $n=2$.

3) We replaced Supplementary Information with Expanded View (EV) Figures and Tables that are collapsible/expandable online. A maximum of 5 EV Figures can be typeset. EV Figures should be cited as 'Figure EV1, Figure EV2' etc... in the text and their respective legends should be included in the main text after the legends of regular figures.

5) a complete author checklist, which you can download from our author guidelines <https://www.embopress.org/page/journal/14693178/authorguide>. Please insert information in the checklist that is also reflected in the manuscript. The completed author checklist will also be part of the RPF.

6) Please note that all corresponding authors are required to supply an ORCID ID for their name upon submission of a revised manuscript (<https://orcid.org/>). Please find instructions on how to link your ORCID ID to your account in our manuscript

tracking system in our Author guidelines

<<https://www.embopress.org/page/journal/14693178/authorguide#authorshipguidelines>>

10) Regarding data quantification (see Figure Legends:

<https://www.embopress.org/page/journal/14693178/authorguide#figureformat>)

12) All Materials and Methods need to be described in the main text using our 'Structured Methods' format, which is required for all research articles. According to this format, the Methods section includes a separate Reagents and Tools Table file (listing key reagents, experimental models, software and relevant equipment and including their sources and relevant identifiers) and a Methods and Protocols section describing the methods using a step-by-step protocol format. The aim is to facilitate adoption of the methodologies across labs. More information on how to adhere to this format as well as a downloadable template (.docx) for the Reagents and Tools Table can be found in our author guidelines:

An example of a Method paper with Structured Methods can be found here: <https://www.embopress.org/doi/full/10.1038/s44320-024-00037-6#sec-4>

As part of the EMBO publication's Transparent Editorial Process, EMBO reports publishes online a Review Process File (RPF) to accompany accepted manuscripts. This File will be published in conjunction with your paper and will include the referee

reports, your point-by-point response and all pertinent correspondence relating to the manuscript.

I look forward to seeing a revised form of your manuscript when it is ready.

Referee #1:

The study by Park et al. presents an interesting and potentially important mechanism linking neuronal activity to local protein synthesis and axonal regeneration. The authors show that activation of NMDARs triggers calcium influx, which activates calpain 1 and leads to cleavage of the RNA-binding protein G3BP1 at its proline-rich IDR2 domain. This cleavage causes disassembly of G3BP1-containing ribonucleoprotein granules, stimulates mTOR-dependent local translation, and promotes axonal regrowth. The findings suggest a new way that synaptic signals can reshape local proteomes through controlled proteolysis, which could have broad implications for neuronal plasticity and repair. That said, several points need to be addressed before the conclusions can be fully supported:

1. Many key datasets (e.g., Figs. 1A, 1D-E, 2H, 3E, 4C, 5B, and 6A) lack proper statistical analysis, and some images (notably Fig. 1B) need clearer presentation. Quantifying G3BP1 intensity in dendrites and axons and normalizing for axonal thickness and branching would strengthen the results.
2. The authors should clarify whether the decrease in insoluble G3BP1 following glutamate stimulation is beneficial or harmful to neurons, especially when compared to sodium arsenite, which increases insoluble G3BP1 and is known to cause stress. Including data on neuronal survival or stress markers would help interpret the functional consequences.
3. The evidence that G3BP1 is cleaved by calpain 1 is convincing, but it remains unclear whether G3BP1 is the main or unique substrate driving local translation and regeneration. Calpain 1 is known to target many neuronal proteins. To establish specificity, the authors should:
 - a. Use calpain 1 and calpain 2 knockdown or knockout neurons to test functional outcomes.
 - b. Introduce a cleavage-resistant G3BP1 mutant to see whether it blocks NMDA-induced granule disassembly, translation, and axonal regeneration.
 - c. Perform a calpain 1 degradome or N-terminomics analysis to identify all substrates cleaved after NMDAR activation, determining whether G3BP1 is a key or one among several relevant targets.
4. The link between G3BP1 cleavage and enhanced translation is inferred but not directly shown. The authors should test whether preventing G3BP1 cleavage, either pharmacologically or genetically, also prevents local puromycin incorporation and mTOR activation, proving that cleavage is required for translational derepression.
5. mTOR activity is inferred mainly from imaging. Western blot analyses of mTOR and its downstream under NMDA, MDL-28170, or Torin 1 treatment would provide stronger evidence for mTOR involvement.
6. To confirm that the observed translation truly occurs locally, experiments using compartmentalized neuronal cultures that physically separate axons from cell bodies should be performed. Comparing translation between compartments would clarify the spatial specificity of the effect.
7. The presence of G3BP1 truncation in the basal condition in Figure 6A contradicts earlier results showing no truncation at baseline (Figure 2). The authors should explain this discrepancy, whether due to developmental stage, injury model, or experimental variability.
8. The findings should be extended to an *in vivo* setting. Testing whether the NMDAR-calpain 1-G3BP1-mTOR pathway functions during regeneration in a nerve injury or brain lesion model would make the conclusions much more compelling.
9. A global proteomic analysis identifying other calpain 1 substrates and newly synthesized proteins following NMDAR activation would greatly strengthen the study. This would clarify whether G3BP1 is the central effector or part of a wider proteolytic network that remodels the neuronal proteome to support local translation and regeneration.

Referee #2:

The authors report stimulus dependent proteolysis of the stress granule protein G3BP1 in response to NMDA receptor activation

in axons and dendrites. The cleavage disassembles G3BP1 granules (presumably stress granules) and allows activation of mTOR-dependent translation. This is an intriguing mechanism to drive stimulus dependent local translation, similar to G3BP1's post-translational modifications that the authors reference. I think they have definitively shown calpain-dependent cleavage of G3BP1 through NMDAR activation, but there are a number of weaknesses that detract from my opinion of the manuscript in its present form. On the one hand, I think they have over interpreted some findings. On the other hand, there are sufficient uncertainties in several of the figures that raise concerns on interpretation. Also some points missing from the discussion that would strengthen the manuscript.

Major issues:

1. The focus on axons is a bit surprising given the obvious roles of NMDAR in post-synaptic functions. The authors should introduce the logic for this sooner in the manuscript. Also, are the cultures analyzed at point when pre-synaptic NMDAR signaling is prominent?
2. The authors provide clear blots for Figures 1 & 2 where G3BP1 cleavage is undeniable with KCl & glutamate stimuli but not arsenite. But the blots for Figures 3B-E, 4 & 6 have many extraneous bands. The authors need to better annotate these panels. Many of the blots seem overexposed so major bands cannot be distinguished. A minor issue is the migration of full length G3BP1 - predicted MW is ~ 52 kDa but the authors report something closer to 70 kDa (a sentence explaining this would help the reader). Similarly, it would be beneficial to know the epitope for the G3BP1 antibody (presumably N-terminal oriented) and if the authors pick up the ~25 kDa band from Figure 3D with a C-terminal epitope (if such antibody is available). A related point is what have the authors considered what the truncated peptides for G3BP1 might be doing? Figure 4 starts to get at this by examining domains, but the authors have not considered whether the cleaved G3BP1 fragments have activity. The N terminal fragment could likely still dimerize with uncleaved G3BP1, possibly preventing its LLPS. The C-terminal fragment might have RNA binding activity on its own. I think these points are worth mentioning in the discussion.
3. Figure 4 blots are particularly confusing, at least for this reviewer. As noted in # 1 above, there are many bands for panels B, D, E, F, & G - I am forced to take the author interpretation of the findings rather than be guided through that interpretation. The authors must annotate these better for the readers' (& reviewer's) interpretation. The authors' interpret this figure as Calpain being constitutively bound to G3BP1. First, I do not see evidence of this. Second, these data are limited to overexpression in HEK cells that the authors infer applies to endogenous G3BP1-Calpain in axons and dendrites. The remainder of the data prove that calpain is involved, but do not support constitutive binding. Studies in primary neurons with the endogenous proteins are need to make that claim. This could be done by co-immunoprecipitations or colocalizations (better if both).
4. The puromycinylation studies in Figure 5 show compelling differences. But the blots in panel C are woefully overexposed. I can see the logic of overexposing the -NMDA lanes, but that is at the cost of seeing definitive bands in those lanes as well as the +NMDA lanes. Perhaps the authors show 2 exposures?
5. Though the effect of MDL281270 & EGTA are clear in Figure 6E, mTOR inhibitor effect is only partial. I think the authors need to quantify this finding. Further, they need to show to what extent Torin1 is blocking activity in these neurites. Previous work from Sahoo et al., 2018, Nat Commun showed that depletion of G3BP1 promotes axon growth - it would be intriguing to see if that is the case in this system (at minimal, the authors should discuss this as well as the C elegans TIAR1 depletion reported by Andrusiak et al, 2010, Neuron - both support the authors' conclusions on G3BP1 role).

Minor points:

- i. Figure 1C-E would benefit from images of axons and dendrites for AsNO₄ treatments. Also where along the axon & dendrite were the images from D & E taken. Figure 2H would benefit from similar images of dendrites.
- ii. For Figure 5E & G, the authors should designate whether the RPM values for MDL28170 + NMDA are significantly different than basal & NMDA. Also the authors need to define what statistical test was used and N in the figure legend.

Referee #3:

This manuscript reports some new and interesting results linking NMDA receptor activation to calpain-1 stimulation, the resulting cleavage of G3BP1 and the dissolution of ribonucleoprotein granules and the regulation of local protein synthesis. While the results are relatively convincing the manuscript has a number of significant problems.

1. All the western blots indicate that there are 2 bands for native G3BP1. The authors need to explain what these 2 bands represent and whether they are equally susceptible to calpain-1-mediated cleavage.
2. A major issue is that most results are qualitative and there are no quantifications, at least for all the western blots.

3. It has been repeatedly shown that application of glutamate to cultured neurons does not activate the glutamate receptors but rather results in glutamate uptake. The significance of the results obtained with glutamate application needs to be discussed.
4. Most of the results are illustrating axonal localization of the proposed mechanism. The authors do indicate that this also takes place in the dendrites, but they need a better illustration to convince the reader that this is actually true.
5. All the experiments are conducted in cultured neurons. It would be important to perform experiments indicating that this mechanism is also present in adult rats.
6. Many figures need to be fixed as the labels above the western blots appear to be shifted.

Referee #1:

The study by Park et al. presents an interesting and potentially important mechanism linking neuronal activity to local protein synthesis and axonal regeneration. The authors show that activation of NMDARs triggers calcium influx, which activates calpain 1 and leads to cleavage of the RNA-binding protein G3BP1 at its proline-rich IDR2 domain. This cleavage causes disassembly of G3BP1-containing ribonucleoprotein granules, stimulates mTOR-dependent local translation, and promotes axonal regrowth. The findings suggest a new way that synaptic signals can reshape local proteomes through controlled proteolysis, which could have broad implications for neuronal plasticity and repair. That said, several points need to be addressed before the conclusions can be fully supported:

We sincerely thank the reviewer for the clear and thoughtful summary of our work. We are pleased that the study's central findings are considered interesting and potentially important, and we appreciate the reviewer's recognition of the broader implications of our findings.

In the revised manuscript, we have carefully addressed all of the points raised below to strengthen our conclusions and improve the clarity and rigor of the study. We believe these revisions have substantially enhanced the quality of the manuscript.

1. Many key datasets (e.g., Figs. 1A, 1D-E, 2H, 3E, 4C, 5B, and 6A) lack proper statistical analysis, and some images (notably Fig. 1B) need clearer presentation. Quantifying G3BP1 intensity in dendrites and axons and normalizing for axonal thickness and branching would strengthen the results.

Response: We thank the reviewer for this important comment. In the revised manuscript, we have performed appropriate statistical analyses for the relevant datasets and included the corresponding quantifications in the following revised figure panels: Fig. 1B (for 1A), 1F and 1H (for 1D-E), 2J (for 2H), 3F (for 3E), 4D (for 4C), 5C (for 5B), and 6B (for 6A). Notably, for G3BP1 intensity quantification, we consistently selected first- or second-order branches for dendrites, and analyzed regions of comparable thickness in both dendrites and axons, based on MAP2- or Tau-positive immunostaining, respectively.

2. The authors should clarify whether the decrease in insoluble G3BP1 following glutamate stimulation is beneficial or harmful to neurons, especially when compared to sodium arsenite, which increases insoluble G3BP1 and is known to cause stress. Including data on neuronal survival or stress markers would help interpret the functional consequences.

Response: We thank the reviewer for raising this important point. To assess whether the observed decrease in insoluble G3BP1 following NMDA stimulation is associated with neuronal stress or toxicity, we compared three conditions treated for 2 hours: NMDA (which induces granule disassembly), NMDA with calpain inhibition (which prevents disassembly), and sodium arsenite (which promotes granule assembly). Using an LDH cytotoxicity assay, we found no significant differences in neuronal viability across these conditions. In addition, we assessed

stress markers including BiP/GRP78 and cleaved caspase-3, and observed minimal changes among the groups (Rebuttal Figs. 1A,B). These results suggest that, under the conditions used in our study, neither NMDA-induced granule disassembly nor arsenite-induced granule formation exerts overtly beneficial or detrimental effects on neuronal viability. We have included the neuronal viability data in the revised Fig. EV2.

3. The evidence that G3BP1 is cleaved by calpain 1 is convincing, but it remains unclear whether G3BP1 is the main or unique substrate driving local translation and regeneration. Calpain 1 is known to target many neuronal proteins. To establish specificity, the authors should:

a. Use calpain 1 and calpain 2 knockdown or knockout neurons to test functional outcomes.

Response: We successfully achieved calpain 2 knockdown in neurons using shRNA; however, despite multiple independent attempts using five different target sequences, we were unable to achieve effective knockdown of calpain 1.

In injury-induced axonal regeneration assays, calpain 2 knockdown alone did not significantly affect axonal regrowth (revised Figs. EV4A–C). In contrast, treatment with MDL-28170, which inhibits both calpain 1 and calpain 2, markedly reduced axonal regeneration under conditions of calpain 2 knockdown (revised Figs. EV4A–C), indicating that the regenerative process is primarily dependent on calpain 1 activity.

Together with our biochemical data showing that calpain 2 does not cleave G3BP1 (Figs. 3B, 3C, and 3E), these results suggest that calpain 2 does not contribute to G3BP1-mediated regeneration and may instead play a distinct or opposing role relative to calpain 1, supporting the isoform-specific functions of calpains in neuronal regeneration.

b. Introduce a cleavage-resistant G3BP1 mutant to see whether it blocks NMDA-induced granule disassembly, translation, and axonal regeneration.

Response: We expressed a cleavage-resistant G3BP1 Δ PxxP mutant, or wild-type G3BP1, using lentiviral transduction in primary cortical neurons and examined NMDA-induced local translation and axonal regeneration. Following NMDA stimulation, neurons expressing G3BP1 Δ PxxP did not show increased RPM signals, in contrast to neurons expressing wild-type G3BP1, indicating that prevention of calpain-mediated G3BP1 cleavage blocks NMDAR-dependent enhancement of local translation (revised Figs. 5G,H). Furthermore, expression of the G3BP1 Δ PxxP mutant markedly reduced axonal regeneration in the injury model (revised Figs. 6H,I). Together, these results demonstrate that calpain 1-mediated cleavage of G3BP1 is required for NMDAR-dependent translational derepression and axonal regeneration, supporting the specificity of G3BP1 as a key functional substrate of calpain 1 in this pathway.

c. Perform a calpain 1 degradome or N-terminomics analysis to identify all substrates cleaved after NMDAR activation, determining whether G3BP1 is a key or one among several relevant targets.

Response: We fully acknowledge the value of degradome or N-terminomics approaches to systematically identify calpain 1 substrates following NMDAR activation. However, such large-scale proteomic analyses are currently beyond the technical and resource capacities of our laboratory. We consider this an important avenue for future research that could further define the scope of activity-dependent proteolytic remodeling in neurons.

We have added the following statement to the Discussion section of the revised manuscript: *“Furthermore, it would be highly informative to perform a mass spectrometry-based approach to identify the full spectrum of calpain 1 substrates following NMDAR activation. For example, a recent study successfully mapped calpain 2 substrates (Kaplan et al., Protein Science 2025; 34(5):e70144), and similar efforts targeting calpain 1 could provide invaluable insights into the broader proteolytic signaling network governed by calpain 1.”*

4. The link between G3BP1 cleavage and enhanced translation is inferred but not directly shown. The authors should test whether preventing G3BP1 cleavage, either pharmacologically or genetically, also prevents local puromycin incorporation and mTOR activation, proving that cleavage is required for translational derepression.

Response: We have already addressed this concern by performing the ribopuromylation (RPM) assay together with G3BP1 immunostaining (Fig. 5E-F). These experiments showed that local puromycin incorporation occurs in close proximity to G3BP1 granules under conditions that promote G3BP1 cleavage (i.e., NMDAR stimulation), and this signal is markedly reduced following pharmacological inhibition with MDL-28170. These findings suggest that G3BP1 cleavage is required for the derepression of local translation.

To further strengthen this causal link, we examined NMDAR-induced local translation using a cleavage-resistant G3BP1 Δ PxxP mutant. As described in our response to Comment #3b, NMDA-induced increases in the RPM signal were not observed in neurons expressing the Δ PxxP mutant (revised Figs. 5G,H). This result indicates that G3BP1 cleavage is necessary for local puromycin incorporation, thereby supporting the conclusion that cleavage of G3BP1 is required for translational derepression.

5. mTOR activity is inferred mainly from imaging. Western blot analyses of mTOR and its downstream under NMDA, MDL-28170, or Torin 1 treatment would provide stronger evidence for mTOR involvement.

Response: Our data indicate that axonal local translation of mTOR is increased upon NMDA treatment, as demonstrated by Puro-PLA imaging (Figs. 5I,J), whereas global protein synthesis is reduced (Fig. 5D). To further examine mTOR pathway activation, we performed Western blot analyses of mTOR and its downstream effectors, including phosphorylated mTOR, p70S6K and 4E-BP, in neurite-enriched fractions of primary hippocampal neurons (see also our response to Comment #6). However, we did not observe a measurable increase in mTOR activity using this approach (revised Fig. EV3A,B). We propose that these results may reflect the presence of a substantial pool of pre-existing mTOR protein, which could obscure subtle changes in newly synthesized or activated mTOR. While Puro-PLA imaging can selectively detect nascent mTOR

synthesis, bulk Western blotting lacks the spatial and temporal resolution to distinguish these localized, activity-dependent changes from the pre-existing mTOR pool.

6. To confirm that the observed translation truly occurs locally, experiments using compartmentalized neuronal cultures that physically separate axons from cell bodies should be performed. Comparing translation between compartments would clarify the spatial specificity of the effect.

Response: To directly confirm the spatial specificity of translation, we cultured primary hippocampal neurons on porous membrane inserts using a Boyden chamber system, which physically separates neurites from cell bodies. We then performed ribopuromycylation (RPM) assays in the presence of emetine and puromycin. NMDA treatment increased local translation in the neurite-enriched fraction, and this effect was reversed by co-treatment with the calpain inhibitor MDL-28170 (revised Figs. 5K,L). These results further support that NMDA-induced local protein synthesis occurs in a calpain-dependent manner within neurites.

7. The presence of G3BP1 truncation in the basal condition in Figure 6A contradicts earlier results showing no truncation at baseline (Figure 2). The authors should explain this discrepancy, whether due to developmental stage, injury model, or experimental variability.

Response: We thank the reviewer for pointing this out. We have repeated the experiment shown in Figure 6A and now provide updated data with quantification in revised Figures 6A and 6B, confirming that G3BP1 truncation is not observed under baseline conditions. We believe the initial discrepancy may have been due to differences in neuronal culture conditions or variability in cell health between preparations, which can affect basal proteolytic activity. This has been clarified in the revised figure with accompanying quantification.

8. The findings should be extended to an in vivo setting. Testing whether the NMDAR-calpain 1-G3BP1-mTOR pathway functions during regeneration in a nerve injury or brain lesion model would make the conclusions much more compelling.

Response: We fully agree with the reviewer that validating our findings in an in vivo injury model would further strengthen the overall impact of the study. However, we believe that the core mechanistic insights regarding the NMDAR-calpain 1-G3BP1-mTOR signaling axis are robustly supported by our current data in primary neuronal cultures. At this time, in vivo nerve injury or brain lesion models are not yet established in our laboratory, and implementing such experiments would require significant time and resources, which could considerably delay the revision process. We respectfully propose that in vivo validation represents an important future direction to build upon the present findings.

We have added the following statement to the Discussion section of the revised manuscript: *“Nonetheless, as the current study is based on primary cultured neuronal models, future investigations using in vivo injury models will be essential to further elucidate the physiological relevance and functional significance of the signaling axis in neural repair.”*

9. A global proteomic analysis identifying other calpain 1 substrates and newly synthesized proteins following NMDAR activation would greatly strengthen the study. This would clarify whether G3BP1 is the central effector or part of a wider proteolytic network that remodels the neuronal proteome to support local translation and regeneration.

Response: We fully agree with the reviewer that a global proteomic analysis would provide valuable insight into the broader proteolytic network regulated by calpain 1. However, we believe that this type of large-scale analysis extends beyond the current scope and primary focus of our study, which aims to elucidate the specific mechanistic pathway linking NMDAR–calpain 1–G3BP1 signaling to mTOR-dependent local translation and axonal regeneration. We consider this an important direction for future research. Please also refer to our response to Comment #3c.

Referee #2:

The authors report stimulus dependent proteolysis of the stress granule protein G3BP1 in response to NMDA receptor activation in axons and dendrites. The cleavage disassembles G3BP1 granules (presumably stress granules) and allows activation of mTOR-dependent translation. This is an intriguing mechanism to drive stimulus dependent local translation, similar to G3BP1's post-translational modifications that the authors reference. I think they have definitively shown calpain-dependent cleavage of G3BP1 through NMDAR activation, but there are a number of weaknesses that detract from my opinion of the manuscript in its present form. On the one hand, I think they have over interpreted some findings. On the other hand, there are sufficient uncertainties in several of the figures that raise concerns on interpretation. Also some points missing from the discussion that would strengthen the manuscript.

We sincerely thank the reviewer for the thoughtful evaluation of our findings regarding NMDAR-dependent, calpain-mediated G3BP1 cleavage and its role in regulating local translation. We are pleased that the reviewer finds our identification of stimulus-dependent mechanism for local translation to be intriguing. We fully acknowledge the reviewer's concerns about potential overinterpretation and several weaknesses in data presentation. In the revised manuscript, we have carefully addressed each point and improved the clarity and rigor of the data and interpretation throughout the revised manuscript.

Major issues:

1. The focus on axons is a bit surprising given the obvious roles of NMDAR in post-synaptic functions. The authors should introduce the logic for this sooner in the manuscript. Also, are the cultures analyzed at point when pre-synaptic NMDAR signaling is prominent?

Response: We have revised the Introduction to clarify the rationale for focusing on axonal NMDARs and have included supporting references. The revised text now reads:

“Although NMDARs are best known for their postsynaptic functions, presynaptic NMDARs (preNMDARs) are also widely expressed at specific synapses throughout the central nervous system (CNS), where preNMDARs have been implicated in the regulation of neurotransmitter release and various forms of synaptic plasticity (Bouvier et al., 2015 Neuroscience; Wong et al., 2021 J Physiol). PreNMDAR expression is particularly enriched during early postnatal development and often includes juvenile subunits such as GluN2B and GluN3A. The low Mg²⁺ sensitivity of preNMDARs suggests that strong stimuli, such as high-frequency firing or axonal injury, may enhance preNMDAR activation and downstream signaling. Notably, a recent study demonstrated that high-frequency neurotransmission in neocortical layer 5 pyramidal neurons is sustained through a preNMDAR-mTOR signaling pathway that controls local protein synthesis (Wong et al., 2024 Neuron).”

In our study, the primary neurons used were mostly mature (DIV14–16), while younger neurons (DIV10–14) were used in the injury model. We believe that these timepoints fall within the developmental window during which presynaptic NMDAR signaling remains functionally relevant. Moreover, strong stimuli such as NMDA treatment or mechanical injury may further potentiate preNMDAR activity even in mature neurons, supporting the observed axonal responses in our experiments.

2. The authors provide clear blots for Figures 1 & 2 where G3BP1 cleavage is undeniable with KCl & glutamate stimuli but not arsenite. But the blots for Figures 3B-E, 4 & 6 have many extraneous bands. The authors need to better annotate these panels. Many of the blots seem overexposed so major bands cannot be distinguished. A minor issue is the migration of full length G3BP1 - predicted MW is ~ 52 kDa but the authors report something closer to 70 kDa (a sentence explaining this would help the reader). Similarly, it would be beneficial to know the epitope for the G3BP1 antibody (presumably N-terminal oriented) and if the authors pick up the ~25 kDa band from Figure 3D with a C-terminal epitope (if such antibody is available).

Response: We thank the reviewer for this constructive comment. We have repeated the experiment originally shown in Figure 3C to reduce overexposure and clarify band patterns, and present the updated data in revised Figure 3C. In addition, we have annotated the relevant bands in revised Figures 3B, 3C, and 3E to facilitate interpretation. As the reviewer noted, overexpression of G3BP1 in heterologous cells often produces additional extraneous bands, which are much less prominent in primary neurons analyzing endogenous G3BP1. We believe these extra bands likely reflect non-physiological or off-target cleavage events under overexpression conditions, and are not observed in the endogenous neuronal context.

Regarding the migration of full-length G3BP1, although the calculated molecular weight of G3BP1 is approximately 52 kDa, the observed molecular weight in neuronal lysates is ~70 kDa, consistent with the manufacturer’s datasheet for this antibody. This apparent shift is likely due to the unique structural and biochemical properties of G3BP1. First, the presence of a highly acidic domain may reduce SDS binding efficiency, decreasing the net negative charge, and resulting in slower migration through the gel. Second, the protein’s intrinsically disordered regions (IDRs), which lack compact tertiary structure, may cause increased resistance during electrophoresis. These features likely contribute to the aberrant migration behavior and apparent higher molecular weight.

The antibody used in this study recognizes an epitope corresponding to amino acids 167–466 of G3BP1 (full-length: 1–466), spanning the acidic, PxxP, and PBD domains, but not the N-terminal NTF2-like domain. Since the cleavage site lies within the PxxP region, this antibody can detect both N-terminal and C-terminal fragments. However, the endogenous C-terminal fragment appears to be highly unstable and was difficult to detect in primary neurons following NMDA treatment, in contrast to clearer detection in heterologous overexpression systems (see also our response to the Comment below).

To help clarify molecular weight discrepancy for readers, we have added the following statement to the first Results section of the revised manuscript: “*Although the calculated molecular weight of G3BP1 is approximately 52 kDa, the observed band in neurons appears at approximately 70 kDa, likely due to unique structural and biochemical features of G3BP1, such as its acidic domain and IDRs, which may alter its migration during SDS-PAGE.*”

A related point is what have the authors considered what the truncated peptides for G3BP1 might be doing? Figure 4 starts to get at this by examining domains, but the authors have not considered whether the cleaved G3BP1 fragments have activity. The N terminal fragment could likely still dimerize with uncleaved G3BP1, possibly preventing its LLPS. The C-terminal fragment might have RNA binding activity on its own. I think these points are worth mentioning in the discussion.

Response: Thank you for this insightful comment. We fully agree with the reviewer that the calpain-cleaved N-terminal fragment of G3BP1 may retain the ability to dimerize with full-length G3BP1, potentially acting in a dominant-negative manner to disrupt LLPS. Additionally, the C-terminal fragment may possess residual RNA-binding activity through its RRM and RG-rich domains. However, in our hands, the endogenous the C-terminal cleavage fragment appeared to be highly unstable and was difficult to detect in primary neurons following NMDA treatment.

We have added the following statement to the Discussion section of the revised manuscript: “*The calpain-cleaved N-terminal fragment of G3BP1 likely retains dimerization capacity with full-length G3BP1, which could interfere with LLPS and stress granule formation. While the C-terminal fragment may retain RNA-binding activity, it appears to be highly unstable and was difficult to detect in neurons.*”

3. Figure 4 blots are particularly confusing, at least for this reviewer. As noted in # 1 above, there are many bands for panels B, D, E, F, & G - I am forced to take the author interpretation of the findings rather than be guided through that interpretation. The authors must annotate these better for the readers' (& reviewer's) interpretation.

The authors' interpret this figure as Calpain being constitutively bound to G3BP1. First, I do not see evidence of this. Second, these data are limited to overexpression in HEK cells that the authors infer applies to endogenous G3BP1-Calpain in axons and dendrites. The remainder of the data prove that calpain is involved, but do not support constitutive binding. Studies in primary neurons with the endogenous proteins are need to make that claim. This could be done by co-immunoprecipitations or colocalizations (better if both).

Response: We thank the reviewer for this valuable comment. We have improved Figure 4C to reduce overexposed and unclear bands and have added annotations to clarify the interpretation of the blots. We believe that the presence of multiple cleavage bands in our overexpression system likely reflects off-target or non-physiological cleavage events, which are not observed in the endogenous neuronal context.

Regarding the claim of constitutive binding between calpain 1 and G3BP1, we agree with the reviewer that our original data using HEK cell overexpression was insufficient to support this conclusion. To address this, we performed co-immunoprecipitation experiments using endogenous proteins from primary hippocampal neuron lysates. These experiments revealed a clear constitutive interaction between endogenous calpain 1 and G3BP1 (revised Fig. 4E). In addition, we observed substantial co-localization of endogenous calpain 1 and G3BP1 in axons by confocal microscopy (revised Fig. 4F). These new data have been incorporated into the revised manuscript and further support our model that calpain 1 is pre-associated with G3BP1 in resting neurons, enabling rapid and spatially restricted proteolysis upon calcium influx.

4. The puromycinylation studies in Figure 5 show compelling differences. But the blots in panel C are woefully overexposed. I can see the logic of overexposing the -NMDA lanes, but that is at the cost of seeing definitive bands in those lanes as well as the +NMDA lanes. Perhaps the authors show 2 exposures?

Response: We thank the reviewer for pointing this out. In the original manuscript, we intentionally presented an overexposed puromycin blot in Figure 5C to highlight the relatively low levels of nascent protein synthesis in the +NMDA condition. We have now included an additional, less-exposed version of the blot in the revised Figure 5D. This allows for a more accurate comparison of puromycin incorporation levels and better reflects the dynamic range of nascent protein synthesis under different treatment conditions.

5. Though the effect of MDL281270 & EGTA are clear in Figure 6E, mTOR inhibitor effect is only partial. I think the authors need to quantify this finding. Further, they need to show to what extent Torin1 is blocking activity in these neurites. Previous work from Sahoo et al., 2018, Nat Commun showed that depletion of G3BP1 promotes axon growth - it would be intriguing to see if that is the case in this system (at minimal, the authors should discuss this as well as the C elegans TIAR1 depletion reported by Andrusiak et al, 2010, Neuron - both support the authors' conclusions on G3BP1 role).

Response: We appreciate the reviewer's valuable feedback. To address the partial effect of Torin1 observed in original Figure 6E, we extended the axonal regeneration assay to 48 hours post-injury, a time point that revealed more pronounced differences across conditions. Quantification of axonal regeneration showed that Torin1 treatment resulted in an approximately 60% reduction in regrowth compared to the control (revised Figs. 6F,G), confirming that mTOR plays a significant role in the regenerative process.

We are also grateful for the suggestion of prior studies highlighting the role of core stress granule proteins in axonal regeneration. In line with these studies, we examined the functional

consequences of blocking G3BP1 cleavage by expressing a calpain-resistant G3BP1 Δ PxxP mutant in G3BP1 knockdown neurons. We found that G3BP1 Δ PxxP mutant markedly reduced axonal regeneration (revised Figs. 6H,I), consistent with the notion that intact G3BP1 granules suppress regenerative capacity.

We have added the following statement to the Discussion section of the revised manuscript: *“Previous studies have demonstrated that SG components can act as negative regulators of axonal regeneration by repressing local protein synthesis through SG-like aggregation. For example, phosphorylation of G3BP1 at Ser149 promotes SG disassembly and enhances local translation of axonal mRNAs (Sahoo et al., 2018; Sahoo et al., 2025). Disrupting G3BP1-mediated granule formation has also been shown to activate axonal translation and accelerate regeneration (Sahoo et al., 2018; Sahoo et al., 2025). Similarly, TIAR-2 facilitates LLPS-driven granule formation and inhibits axon regeneration in C. elegans (Andrusiak et al., 2019). These findings align with our results and support a model in which proteolytic disassembly of G3BP1 granules facilitates the axonal regeneration process.”*

Minor points:

i. Figure 1C-E would benefit from images of axons and dendrites for AsNO₄ treatments. Also where along the axon & dendrite were the images from D & E taken. Figure 2H would benefit from similar images of dendrites.

Response: We have revised Figs. 1C–E (now shown as revised Figs. 1D–H) by adding representative images of both axons and dendrites under all previous treatment conditions, including the arsenite condition. Additionally, we have included dendritic images in the revised Fig. 2H as suggested.

ii. For Figure 5E & G, the authors should designate whether the RPM values for MDL28170 + NMDA are significantly different than basal & NMDA. Also the authors need to define what statistical test was used and N in the figure legend.

Response: We have statistically analyzed the RPM values for the MDL28170 + NMDA condition in comparison to both the basal and NMDA-only groups using one-way ANOVA followed by Tukey’s post hoc test. The significance levels and N values have been added to revised legends of Figures 5F and 5J, corresponding to the original Figures 5E and 5G, respectively.

Referee #3:

This manuscript reports some new and interesting results linking NMDA receptor activation to calpain-1 stimulation, the resulting cleavage of G3BP1 and the dissolution of ribonucleoprotein granules and the regulation of local protein synthesis. While the results are relatively convincing the manuscript has a number of significant problems.

We sincerely appreciate the reviewer's thoughtful comments on our manuscript. We are pleased that the reviewer finds our research findings interesting and relatively convincing. We fully acknowledge the reviewer's concerns regarding certain aspects of the study. In the revised manuscript, we have carefully addressed each specific point and made substantial revisions to improve the clarity, rigor, and overall quality of the work. We believe these changes significantly strengthen the manuscript.

1. All the western blots indicate that there are 2 bands for native G3BP1. The authors need to explain what these 2 bands represent and whether they are equally susceptible to calpain-1-mediated cleavage.

Response: We thank the reviewer for raising this important point. We hypothesized that the upper band of native G3BP1 might reflect post-translational modifications (PTMs) of G3BP1. To test this possibility, we treated neuronal lysates with lambda phosphatase for up to 24 hours to remove phosphate groups from serine, threonine, and tyrosine residues. However, the two bands remained unchanged following treatment (revised Fig. EV1), suggesting that the upper band is not derived from phosphorylation-dependent PTMs. These findings raise the possibility that the two bands represent alternative transcript isoforms of G3BP1. Additionally, since both the intensities of bands were similarly reduced following NMDA treatment, they appear to be equally susceptible to calpain 1-mediated cleavage.

2. A major issue is that most results are qualitative and there are no quantifications, at least for all the western blots.

Response: We appreciate the reviewer's comment. In the revised manuscript, we have provided quantification for all Western blots. Corresponding statistical analyses have been included in the revised figure legends. (See also our response to Reviewer 1, Comment #1.)

3. It has been repeatedly shown that application of glutamate to cultured neurons does not activate the glutamate receptors but rather results in glutamate uptake. The significance of the results obtained with glutamate application needs to be discussed.

Response: We thank the reviewer for raising this important point regarding the potential confounding effects of glutamate uptake via excitatory amino acid transporters (EAATs) in cultured neurons. Indeed, neurons express both EAAT2 and EAAT3 (Furuta et al., 1997; Mennerick et al., 1998; Chen et al., 2003), and EAATs contribute not only to glutamate clearance but also to neuronal metabolism (Magi et al., 2019). Moreover, under certain conditions, EAAT-mediated uptake can influence receptor activation (Diamond et al., 2001).

However, several lines of evidence suggest that the effects observed in our study are primarily due to glutamate receptor activation, rather than EAAT-mediated uptake. First, our primary neuronal cultures are largely astrocyte-free, containing only a minimal glial component. Under such conditions, bath application of glutamate at 100 μ M falls well within the concentration range known to reliably activate ionotropic glutamate receptors (Rosenberg and Aizenman, 1989;

Rosenberg et al., 1992). Second, we directly confirmed that bath-applied glutamate evokes robust whole-cell inward currents in our cultured neurons, and these responses are fully blocked by ionotropic glutamate receptor antagonists (see Rebuttal Figure 2). This supports that glutamate application activates functional receptors under our experimental conditions. Third, the molecular changes induced by L-glutamate treatment in our study were faithfully reproduced by direct NMDA treatment and were abolished by co-treatment with the NMDAR antagonist D-APV, further supporting a receptor-mediated mechanism. Lastly, although neurons do express EAATs, their uptake capacity is markedly lower than that of astrocytes, based on previous reports showing that astrocyte-poor cultures are ~100-fold more susceptible to NMDA receptor-mediated excitotoxicity than neuron-astrocyte co-cultures (Rosenberg and Aizenman, 1989; Rosenberg et al., 1992).

Taken together, these findings strongly indicate that the effects we observe with glutamate application arise from glutamate receptor activation rather than uptake mechanisms.

4. Most of the results are illustrating axonal localization of the proposed mechanism. The authors do indicate that this also takes place in the dendrites, but they need a better illustration to convince the reader that this is actually true.

Response: We have now included representative images of dendrites as well as axons in the revised figures (Fig. 1E and Fig. 2H) to better illustrate that the proposed mechanism also occurs in dendrites. (Please also see our response to Reviewer 2, Minor point #i.)

5. All the experiments are conducted in cultured neurons. It would be important to perform experiments indicating that this mechanism is also present in adult rats.

Response: We agree with the reviewer that in vivo validation would significantly strengthen our conclusions. However, the establishment and optimization of in vivo CNS injury models are currently beyond the scope of this study and would require considerable time and resources. We have revised the Discussion section to clearly acknowledge this limitation as follows:

“Nonetheless, as the current study is based on primary cultured neuronal models, future investigations using in vivo injury models will be essential to further elucidate the physiological relevance and functional significance of the signaling axis in neural repair.”

6. Many figures need to be fixed as the labels above the western blots appear to be shifted.

Response: We appreciate the reviewer’s attention to detail. We found that angled text labels in particular appeared shifted in some figure panels. We have carefully reviewed and corrected all misaligned labels in the revised figures to ensure accurate and clear presentation.

Rebuttal Figure 1

Rebuttal Figure 1. Two-hour stimulation does not significantly affect neuronal survival or stress marker expression.

(A) Neuronal survival assessed by LDH release assay following 2 h treatment of DIV14 primary hippocampal neurons with 50 μ M NMDA, 50 μ M NMDA plus 5 μ M MDL-28170, or 200 μ M sodium arsenite.

(B) Western blot analysis of stress- and apoptosis-related markers under the same treatment conditions.

Rebuttal Figure 2

Rebuttal Figure 2. Glutamate-evoked whole-cell current in a cultured hippocampal neuron.

A DIV 16 hippocampal neuron was voltage-clamped at -70 mV, and whole-cell current was recorded in the presence of 1 μ M tetrodotoxin (TTX) and 100 μ M picrotoxin (PiTX). The perfusate was sequentially switched from recording solution (with TTX and PiTX) to recording solution containing 100 μ M L-glutamate, and finally to the same solution supplemented with 50 μ M NBQX and 10 μ M MK-801. Bath-applied glutamate elicited a robust inward current that was completely abolished by the ionotropic glutamate receptor antagonist cocktail.

References

- Chen Y., Swanson R. (2003) The glutamate transporters EAAT2 and EAAT3 mediate cysteine uptake in cortical neuron cultures, *J Neurochem*
- Diamond J. (2001) Neuronal Glutamate Transporters Limit Activation of NMDA Receptors by Neurotransmitter Spillover on CA1 Pyramidal Cells, *J Neurosci*
- Furuta A., et al. (1997) Glutamate transporter protein subtypes are expressed differentially during rat CNS development, *J Neurosci*
- Magi S., et al. (2019) Excitatory Amino Acid Transporters (EAATs): Glutamate Transport and Beyond, *Int J Mol Sci*
- Mennerick S., et al. (1998) Neuronal expression of the glutamate transporter GLT-1 in hippocampal microcultures, *J Neurosci*
- Rosenberg P., Aizenman E. (1989) Hundred-fold increase in neuronal vulnerability to glutamate toxicity in astrocyte-poor cultures of rat cerebral cortex, *Neurosci Lett*
- Rosenberg P., et al. (1992) Glutamate uptake disguises neurotoxic potency of glutamate agonists in cerebral cortex in dissociated cell culture, *J Neurosci*

Dear Prof. Suh,

Thank you for the submission of your revised manuscript. We have now received the enclosed reports from the referees and I am happy to say that both support its publication now. Only a few editorial requests will need to be addressed before we can proceed with the official acceptance of your manuscript:

- Please move the Data Availability Statement to before the Acknowledgments.
- The author credits need to be removed from the ms file. All credits need to be entered during online ms submission.
- In the author checklist, all questions regarding statistics need to be answered. Please send us a new, completed checklist.

* Figure Legends - Comments *

- Please note that the figure 1B is mislabeled as figure 1C in the manuscript. This needs to be rectified.
- Please note that the exact p values are not provided in the legends of figures 1B, F, H; 2J, 3F, 4D, 6B, G, I; EV2, please provide exact values as reasonable.
- Please note that the error bars are not defined in the legends of figures 1B, F, H; 2J, 3F, 4D, 5C, F, H, J, L; 6B, G, I; EV2, EV3 B, EV4 C

I would like to suggest to remove the very last part of the last sentence in the abstract (with potential implications for therapeutic intervention in brain injury) because your study does not cover therapy.

I slightly shortened the short summary for our website and combined the second and third bullet point. Do you agree with this:

NMDAR activation induces calpain 1-mediated cleavage of G3BP1, leading to G3BP1 granule disassembly and mTOR-dependent local translation in axons, promoting axonal regeneration.

· Calpain 1-mediated G3BP1 cleavage leads to G3BP1 granule disassembly and stimulates local translation in axons, which is mTOR-dependent

The synopsis image you sent mentions "nerve regeneration". It might be better to replace this with "axon regeneration".

Referee #1:

The authors answered my concerns

Referee #3:

The authors have appropriately revised their manuscript to address all the points raised by the reviewers. It is now suitable for publication.

Point-by-point reply to the reviewers' comments

Paper number: EMBOR-2025-62794V2

Title: Proteolytic cleavage of G3BP1 by calpain 1 couples NMDAR activation to mTOR-dependent local translation

=====

Thank you for the submission of your revised manuscript. We have now received the enclosed reports from the referees and I am happy to say that both support its publication now. Only a few editorial requests will need to be addressed before we can proceed with the official acceptance of your manuscript:

We sincerely thank the Editor and reviewers for their positive evaluation of our study. We have carefully checked the manuscript and addressed all editorial requests as detailed below.

- Please move the Data Availability Statement to before the Acknowledgments.

We have moved the Data Availability Statement to before the Acknowledgments.

- The author credits need to be removed from the ms file. All credits need to be entered during online ms submission.

We have removed the author contribution section from the manuscript.

- In the author checklist, all questions regarding statistics need to be answered. Please send us a new, completed checklist.

We have revised the author checklist accordingly. To complete the checklist requirements, we also revised the *Quantification and statistical analysis* section in the Methods. During this revision, we noticed that one dataset (Figure 5F) did not follow a normal distribution; therefore, we changed the statistical analysis for this dataset to the Kruskal–Wallis test followed by Dunn's post hoc test. The statistical analyses for the other datasets remain unchanged.

* Figure Legends - Comments *

- Please note that the figure 1B is mislabeled as figure 1C in the manuscript. This needs to be rectified.

We apologize for this typographical error. The labeling has now been corrected.

- Please note that the exact p values are not provided in the legends of figures 1B, F, H; 2J, 3F, 4D, 6B, G, I; EV2, please provide exact values as reasonable.

We have now provided the exact p-values directly on the corresponding figures. Only non-significant p-values remain indicated in the figure legends.

- Please note that the error bars are not defined in the legends of figures 1B, F, H; 2J, 3F, 4D, 5C, F, H, J, L; 6B, G, I; EV2, EV3 B, EV4 C

All error bars represent mean \pm SEM. This information has now been clearly defined in the

corresponding figure legends.

I would like to suggest to remove the very last part of the last sentence in the abstract (with potential implications for therapeutic intervention in brain injury) because your study does not cover therapy.

We fully agree with your suggestion. We have removed the last part of the sentence referring to therapeutic implications.

I slightly shortened the short summary for our website and combined the second and third bullet point. Do you agree with this:

NMDAR activation induces calpain 1-mediated cleavage of G3BP1, leading to G3BP1 granule disassembly and mTOR-dependent local translation in axons, promoting axonal regeneration.

- Calpain 1-mediated G3BP1 cleavage leads to G3BP1 granule disassembly and stimulates local translation in axons, which is mTOR-dependent

These revisions are excellent. We have updated the synopsis summary accordingly.

The synopsis image you sent mentions "nerve regeneration". It might be better to replace this with "axon regeneration".

We have replaced "nerve regeneration" with "axon regeneration" in the synopsis image.

Referee #1:

The authors answered my concerns

Referee #3:

The authors have appropriately revised their manuscript to address all the points raised by the reviewers. It is now suitable for publication.

Prof. Young Ho Suh
Seoul National University College of Medicine
Department of Biomedical Sciences
Seoul 03080
Korea, Republic of

Dear Prof. Suh,

I am very pleased to accept your manuscript for publication in the next available issue of EMBO reports. Thank you for your contribution to our journal.

You may qualify for financial assistance for your publication charges - either via a Springer Nature fully open access agreement or an EMBO initiative. Check your eligibility: <https://link.springer.com/journal/44319/how-to-publish-with-us>

>>> Please note that it is EMBO Reports policy for the transcript of the editorial process (containing referee reports and your response letter) to be published as an online supplement to each paper. If you do NOT want this, you will need to inform the Editorial Office via email immediately. More information is available here: <https://link.springer.com/partners/embo-press/editorial-policies#Peer%20review>